# Semantic novelty modulates neural responses to visual change across the human brain

Maximilian Nentwich [1] ✉, Marcin Leszczynski [2,3,4], Brian E. Russ [3,5,6], Lukas Hirsch[1], Noah Markowitz[7], Kaustubh Sapru[1], Charles E. Schroeder[2,3], Ashesh D. Mehta[7,8], Stephan Bickel[3,7,8] & Lucas C. Parra [1] ✉

Our continuous visual experience in daily life is dominated by change. Previous research has focused on visual change due to stimulus motion, eye movements or unfolding events, but not their combined impact across the brain, or their interactions with semantic novelty. We investigate the neural responses to these sources of novelty during film viewing. We analyzed intracranial recordings in humans across 6328 electrodes from 23 individuals. Responses associated with saccades and film cuts were dominant across the entire brain. Film cuts at semantic event boundaries were particularly effective in the temporal and medial temporal lobe. Saccades to visual targets with high visual novelty were also associated with strong neural responses. Specific locations in higher-order association areas showed selectivity to either high or low-novelty saccades. We conclude that neural activity associated with film cuts and eye movements is widespread across the brain and is modulated by semantic novelty.

To study the neural processing of natural visual stimuli, recent work has focused on the experience of watching movies[1–4]. Movies offer a balance between experimental control and the realism of natural environments[1]. Visual dynamics during movie watching are dominated by motion of objects in the scene, the viewer's own eye movements, and film cuts introduced by the film editor to change view angle, but also to transition between events.

Change to the visual input due to stimulus motion in movies is associated with strong neural responses, widespread across the occipital, parietal and temporal lobes[3,4]. It is a more powerful driver of neural responses than low-level features such as luminance and contrast[3,4]. Change to the visual input to the retina is also caused by saccades, the rapid eye movements between fixations, which many studies consider a confound[5,6]. Neural responses to saccades were

thought to be confined to visual processing areas and largely suppressed in higher-order association areas[7]. More recent work, however, indicates that saccades play an important role in the organization of many perceptual and cognitive processes[8–11]. Saccades modulate neural responses across the visual system[8,12,13], medial temporal lobe[14,15], non-visual nuclei of the anterior thalamus[9] and even the auditory cortex[16]. Complicating matters, stimulus motion in movies can also attract saccades[17], and a novel complex visual scene transiently increases saccade rate[18]. In addition, responses to stimulus motion and to saccades are modulated by the semantics of the visual stimulus. For instance, responses to stimulus motion are particularly pronounced when they are associated with 'social' stimuli[4], and saccade-locked potentials in the medial temporal lobe (MTL) can be specific to the target of a saccade, such as faces or objects[10].

[1]Department of Biomedical Engineering, The City College of New York, New York, NY, USA. [2]Departments of Psychiatry and Neurology, Columbia University College of Physicians and Surgeons, New York, NY, USA. [3]Translational Neuroscience Lab Division, Center for Biomedical Imaging and Neuromodulation, Nathan Kline Institute, Orangeburg, NY, USA. [4]Cognitive Science Department, Institute of Philosophy, Jagiellonian University, Kraków, Poland. [5]Nash Family Department of Neuroscience and Friedman Brain Institute, Icahn School of Medicine, New York, NY, USA. [6]Department of Psychiatry, New York University at Langone, New York, NY, USA. [7]The Feinstein Institutes for Medical Research, Northwell Health, Manhasset, NY, USA. [8]Departments of Neurosurgery and Neurology, Zucker School of Medicine at Hofstra/Northwell, Manhasset, NY, USA. ✉e-mail: max.nentwich@gmail.com; parra@ccny.cuny.edu

Besides stimulus motion and saccades, movies also allow us to investigate the processing of narratives, which relies on unique mechanisms in the human brain[19]. The theory of event segmentation proposes that continuous narratives are segmented and remembered as discrete events[20]. Event boundaries, the moments of change between events, are associated with shifts in brain states as well as transient neural responses[6,21–24]. In movies, event boundaries typically coincide with film cuts. Film cuts between events contain semantic changes and are associated with neural activity in higher-order association areas[6,21]. On the other hand, film cuts that maintain continuity (of space, time and action) are mainly associated with changes in low-level visual areas[5,21]. Consistent with event segmentation theory, increased activation in MTL following a film cut is predictive of subsequent recall of the preceding event[25]. Similarly, firing rate of a distinct population of cells in the MTL following film cuts between different clips are predictive of recognition memory[6].

In summary, the main drivers of visual change—stimulus motion, film cuts and saccades—elicit neural activity in various visual and higher-order association areas. However, they have been studied in isolation, so that the relative strength and extent of responses across the brain is not well established. Given that these sources of visual change are correlated it is important to analyze them in combination. Additionally, the effects of event boundaries[20,22,23], social context[4] and faces[15,26] during film suggest a role for novelty at the semantic level. Here, we hypothesized that neural responses associated with stimulus motion, saccades and scene cuts are modulated by semantic novelty in the visual scene. We test this with intracranial EEG recordings that offer high signal-to-noise ratio[27,28] and capture fast responses (<1 s), which may not be detectable with fMRI that is typically used in functional mapping studies. Using a multiple-regression approach we found that neural responses related to stimulus motion are mostly confined to the low-level visual brain areas, while neural responses related to saccades and film cuts are widespread across the whole brain. We refrain from interpreting these responses as being specific to stimulus motion, cuts or saccades, since a variety of features in the film stimuli could be associated with these regressors. To better control for unobserved factors, we then focused on contrasts between high and low semantic novelty while keeping other factors constant. Here we defined low-level semantic novelty in terms of visual features across saccades using deep-networks[29], and on a higher level, in terms of event boundaries judged by human observers across film cuts[20,22,23]. We found that responses to saccades and film cuts are modulated by low and high-level semantic novelty, respectively. Importantly, we found that specific locations in higher-order association areas tend to be modulated by either high or low semantic novelty. Higher-order association areas are defined as those that lie higher on the sensory-to-association hierarchy, as defined by cortical microstructure[30]. The underlying factors that drive these wide ranging and specific responses related to natural dynamic stimuli remain to be untangled. However, we conclude that semantic novelty is an important factor modulating neural responses to film.

## Results

Patients ($N = 23$, Table S1) were implanted with intracranial electrodes for seizure onset localization totaling 6328 contacts with a wide coverage across the whole brain (Fig. 1A). Intracranial electro-encephalography (iEEG) was recorded simultaneously with eye movements while patients watched video clips totaling 43.6 minutes (Fig. 1B). We were interested in neural responses related to the main sources of visual change in movies, motion in the video stimulus, film cuts and eye movements (Fig. 1C). Stimulus motion was quantified as the magnitude of optical flow averaged across the whole screen (Fig. 1C), and captures motion of objects or the whole scene. It is subsequently referred to as motion. Saccades and film cuts were quantified as a series of impulses at the onset of saccades or film cuts,

respectively (Fig. 1C). We observed that film cuts are followed by a significant decrease of saccade frequency after 100 ms and a rebound at 250 ms (Fig. 1D). In addition, motion increases prior to a saccade (Fig. 1E), suggesting that both film cuts and motion drive saccades. Motion also decreases before film cuts (Fig. 1F), as a result of the video editing process (Fig. S1). In total, motion, saccades and cuts clearly interact: saccade frequency increases after film cuts, an increase in motion attracts saccades, and motion slows down before film cuts.

### Neural responses associated with stimulus motion, saccades and film cuts

Based on previous literature we hypothesized that visual change relates to strong and widespread neural responses. To test this we analyzed broad-band high-frequency amplitude (BHA, 70-150 Hz), a signal of dendritic origin that is highly correlated with neuronal firing[31,32]. To increase spatial specificity we performed bipolar re-referencing. All further analysis therefore considers 5378 bipolar channel pairs. We will refer to these channel pairs as channels. To capture neuronal responses, we used a conventional systems identification approach (Fig. S2A)[33]. Specifically, BHA in each channel is treated as the output of a linear system, with motion, saccades and cuts as its input[34]. The resulting impulse responses are often referred to as temporal response functions (TRF) and are obtained for each channel separately.

We test statistical significance separately for responses associated with motion, film cuts and saccades so that each electrode is characterized as responsive to one, several, or none of these stimuli. In each brain area a subset of channels shows statistically significant responses. For example, of the 928 channels in the parietal lobe (Fig. 2A), 238 were responsive to film cuts, 106 to saccades and 65 to motion (FDR corrected with $q < 0.05$, see methods). We analyzed all three sources of change simultaneously to account for the correlation between them and disentangle their contributions to the neural responses[33,34]. For saccade and film cuts this is demonstrated in Fig. S2B. To map responsive channels to the cortical surface we compute a weighted average of the response amplitude in each parcel of the Glasser atlas[35] following a method described by Gao et al.[30] (Fig. 2B).

TRFs vary widely in terms of amplitude, duration and onset across the brain (Figs. 2 and S3–S5). Motion is associated with increased BHA across the brain (Fig. 2A, B and S3A), whereas film cuts and saccades are related to differential increases or decreases in activity, especially beyond sensory areas in the occipital lobe (47% of the responses related to film cuts and 18% of responses related to saccades show a decrease of activity, Figs. 2A, B and S3A). Saccades are associated with a suppression, in particular, prior and during the saccade (in 18% of responsive channels, Figs. 2A and S3A and S5C). The onset of responses associated with film cuts predominantly starts after the cuts, while in the case of motion the onset occurs before the stimulus (Fig. S5). The onset of responses associated with saccades varies widely across the brain with onsets before saccades in some, and after in other areas (Fig. S5). The long-lasting responses of up to 1 s after the saccade are likely associated with processing of the visual stimulus after fixation[36], as opposed to correlated with the saccadic motor command, which takes effect before saccade onset[37]. Consistent with this, we obtain similar responses when using fixation onset as regressor (Fig. S6). However, we prefer to use saccades onset as a marker of visual change, as it is conceptually more closely related to visual change across the retina.

Contrary to our expectation we found the most widespread BHA responses are associated with film cuts followed by saccades and motion (Figs. 2D and S4B). Across the whole brain, the sets of channels responsive to either film cuts, saccades or motion were largely distinct (Fig. 2C). The overlap between responses related to

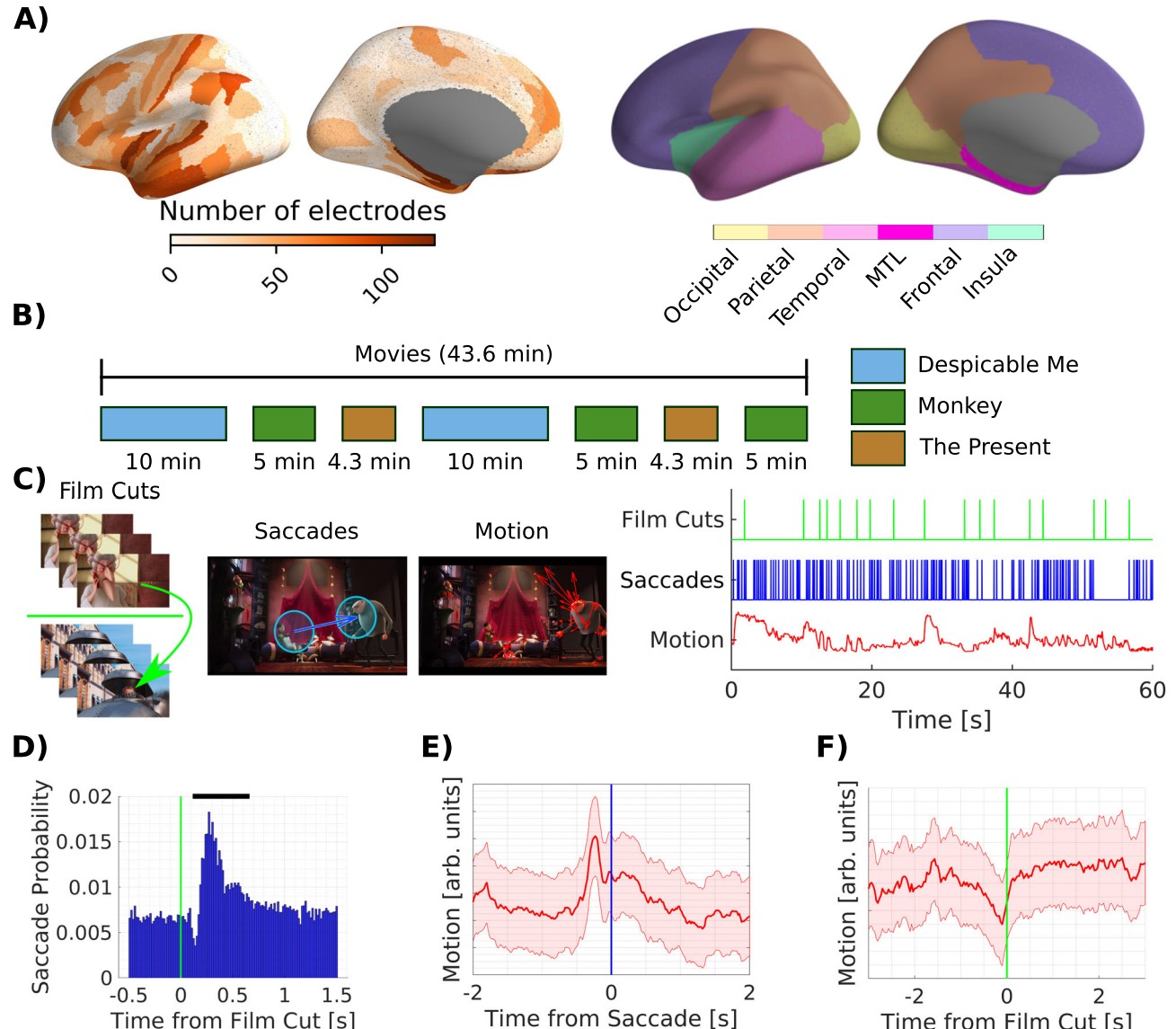

**Fig. 1 | Electrodes coverage, film stimuli and regressors quantifying sources of visual change. A** We analyze neural data in 5378 channels. We localize channels to parcels of the Glasser cortical surface atlas[30,35]. Electrodes cover the whole brain. Density is higher in the temporal and lower in the occipital lobe. Right panel shows the color coding of cortical lobes used in subsequent analysis. The medial temporal lobe (MTL) includes the Amygdala, Hippocampus, entorhinal and para-hippocampal cortex. Contacts in the entorhinal and parahippocampal cortex are excluded from the temporal lobe. **B** Patients watched up to 43.6 min of video clips. Two different 10 min clips of the animated comic 'Despicable Me'[91] were presented, one in English, the other Hungarian. 'The Present' is a short, 4.3 min, animated movie presented twice. 'Monkey' videos are three distinct clips of short scenes from documentaries on macaques presented without sound[4,38]. **C** Sample film cut, with three frames before the cut on top and three frames after the cut on the bottom. Sample saccade on a frame. Blue circles denote the foveal visual field at 5 degree

visual angle. Optical flow vectors (red) on a sample frame. The regressors specifying film cuts and saccade are a series of impulses at the time of the cuts and saccade onset, respectively. The regressor for motion captures the total optic flow in between video frames. "Courtesy of Universal Studios Licensing. © 2010 Universal Animation Studios LLC and Universal City Studios LLLP. All Rights Reserved." **D** Saccade probability as a function of time from film cuts. Black bar indicates time bins with a saccade probability significantly different from the mean (p < 0.001). Significance has been tested against a surrogate distribution of film cuts at random time points. **E** Average motion as a function of saccade onset time. **F** Average motion as a function of time from film cuts across all clips. Cuts tend to follow periods of low motion, an effect mostly driven by 'Despicable Me' (Fig. S1). Shaded area depicts the standard error of the mean. Source data are provided as a Source data file.

saccades and film cuts (41% of saccade-responsive electrodes, Fig. 2C) could be due to unobserved common factors. Surprisingly, the responses related to saccades and film cuts were also stronger than those to motion in areas such as the precuneus and middle temporal area (Fig. S3). These areas have been shown to be important in motion processing[38,39]. However, in the transverse temporal gyrus (Heschl's gyrus) the largest fraction of channels respond to motion (Fig. S3). We refrain from interpreting these responses related to

motion, cuts or saccades as specific to these individual factors. Various features in films could be associated with these regressors. For instance, sound is associated with motion (r = 0.24, p < 10⁻⁴, permutation statistics). This partly explains the responses associated with motion in Heschl's gyrus, which is considered an auditory processing area. To better control for these unobserved factors, we analyzed how each of these factors is modulated by semantic novelty, keeping other confounds constant.

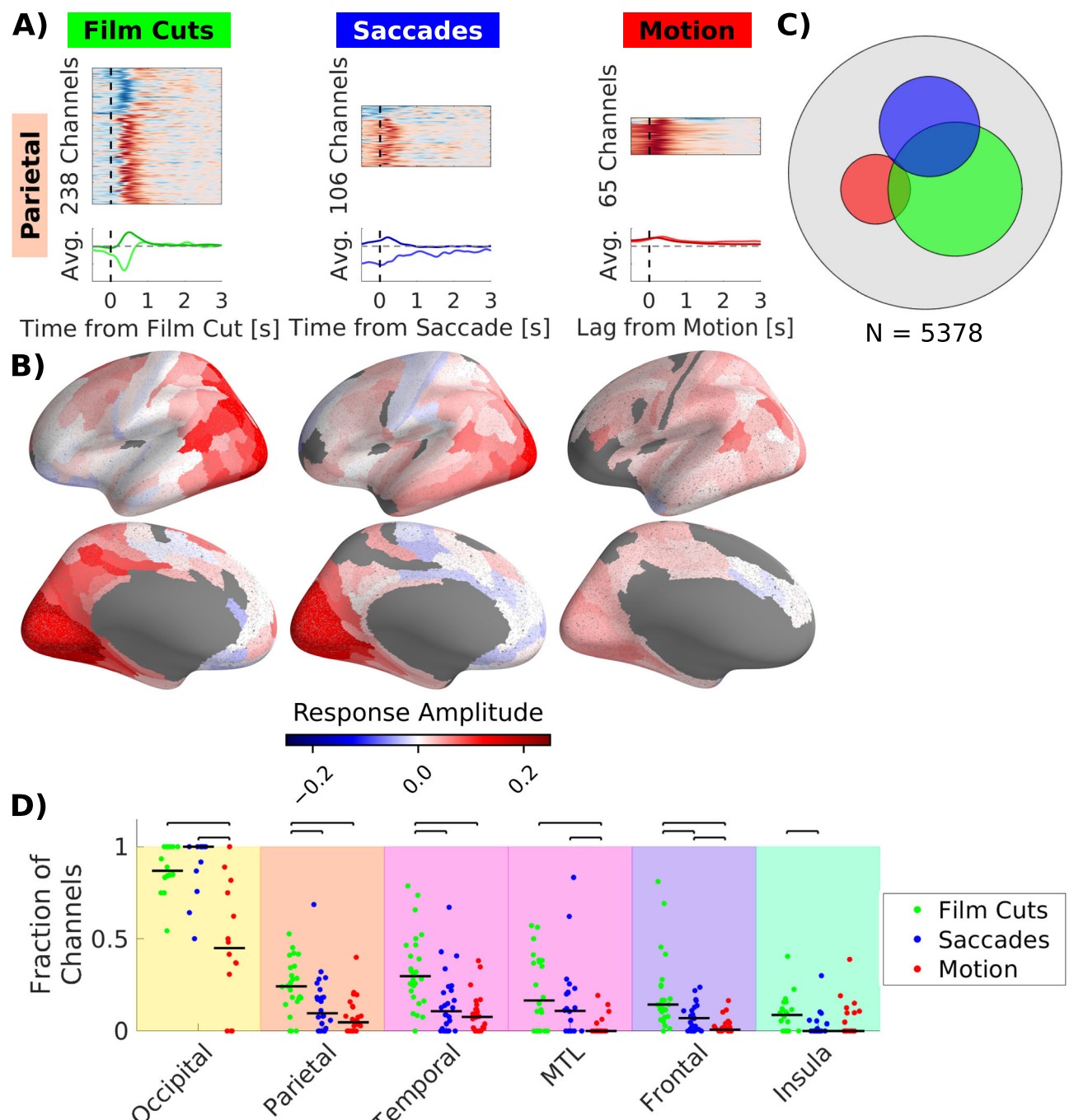

**Fig. 2 | More channels respond to film cuts and saccades than to stimulus motion. A** Top row: Temporal response functions in the parietal lobe for channels with statistically significant response in BHA. Red indicates an increase, blue a decrease in BHA compared to the baseline. The sizes of each image reflect the number of significant channels. Similar TRFs were grouped by clustering TRFs that are highly correlated to each other[86,90]. Bottom row: Average TRF for each cluster, scaled to reflect the relative strength of responses to each regressor within each channel. Time for motion indicates the lag of the neural response in relation to the optical flow signal. **B** Weighted average response amplitude (arb. units) of significant channels in parcels of the Glasser atlas[35]. Red colors indicate an increase of BHA, blue color a decrease of BHA. The number of patients with significant electrodes in each parcel, is plotted in Fig. S4B. **C** Number of significant channels for each regressor indicated as area of each circle. The area of the gray circle indicates the total number of channels. **D** Fraction of channels out of all channels within each brain area with significant response. Each dot shows the fraction of channels with a significant response for each patient. Straight black bars depict the median across patients. Black bars with whiskers above the plot indicate pairwise comparisons with significant differences between the medians (p < 0.05, two-sided Wilcoxon rank sum test, FDR corrected at q = 0.05; Table S2). Background colors correspond to different lobes in Fig. 1A. For results in a more detailed parcellation of the brain see Fig. S3. Source data are provided as a Source data file.

## Cuts at event boundaries are associated with distinct neural responses across the brain

We observed that film cuts are associated with fluctuations in neural activity throughout the brain (Fig. 2). Film cuts cause abrupt changes in various low-level visual features, such as luminance, contrast and color, but for some cuts there are additional changes in semantic information and the narrative. To determine the effect of semantic versus low-level visual changes we divide film cuts into two categories:

"event cuts" and "continuous cuts". (Fig. S7A). To this end, we collected event boundary annotations in a separate population of participants recruited online (N = 180, mean age 27.9 years, age range 19–67 years, 60 female). Participants were instructed to watch the videos and "indicate when […] a meaningful segment has ended by pressing the spacebar". These event boundaries annotations were consistent across participants and with data collected previously[40] (Fig. S7B). For each video, we ranked all cuts by event salience—the number of participants that marked the end of a segment within one second after a cut. The film cuts with large event salience are termed 'event cuts' (using change point detection, see "Methods"). This resulted in 57 event cuts out of a total of 561 cuts (Table S3). Among the cuts with the lowest event salience we selected an equal number of 'continuous cuts' matched in low-level visual features, following[6] (see "Methods"). Continuous cuts lie between narrative event boundaries and are characterized by changes in camera angle or position (Fig. 3C).

We first tested whether event cuts relate to stronger neural responses than continuous cuts. In each channel we fit the TRF identified previously (Fig. 2) to the neural signal after each individual film cut. The factor with which the TRF has to be multiplied to best fit the neural signal, quantifies the amplitude of the response (Fig. S8). We perform a mixed-design ANOVA to test for a fixed effect of event versus continuous cuts and a fixed effect of brain region, while controlling for the random effect of patients (Table S4). We find a significant main effect of cut ($F_{(1,2270)} = 33.8$, $p = 6.9*10^{-9}$) with larger amplitude for event cuts than continuous cuts. Amplitudes also differ significantly across brain regions ($F_{(5,2270)} = 5.75$, $p = 2.1*10^{-5}$). A follow-up mixed-design ANOVA in each brain region with patient as random factor shows that amplitudes are significantly larger for event cuts in the temporal lobe and MTL (Fig. 3A, temporal: $F_{(1,803)} = 48.5$, $p = 7*10^{-12}$; MTL: $F_{(1,95)} = 23.2$, $p = 5.4*10^{-6}$; FDR correction at q = 0.05, full results in Table S5).

We compute the weighted average of the amplitude difference of event versus continuous cuts across all channels in each parcel of the Glasser atlas to visualize this effect on the cortical surface (Fig. 3B)[30,35]. This analysis reveals a complex pattern of brain areas responding preferentially either to event or continuous cuts. We hypothesized that these distinct responses related to event cuts and continuous cuts may be particularly pronounced in higher-order association areas. To explore this possibility, we repeat the analysis but now identify separate TRFs for event cuts and continuous cuts in each channel (Fig. 3C), i.e. using separate regressors indicating each type of cut. We included saccades as a regressor to remove correlated activity associated with saccades following cuts. We obtain similar results when also including motion as a regressor (Fig. S9), or when performing the analysis on different types of videos (Fig. S10). We found channels responsive only to event cuts, channels responsive only to continuous cuts, and channels responsive to both (Fig. S11). We quantify this specificity of responses as the fraction of electrodes responsive to only one type of cut as defined in equation (1). In the occipital lobe specificity is low (Fig. 3D), while in temporal and frontal lobe as well as the MTL it is higher than expected by chance (Table S6). Specificity is mostly driven by channels responsive to event cuts, but not continuous cuts (Fig. S12). In most channels both event cuts and continuous cuts are associated with increased neural activity (Fig. S11). To directly test whether specificity increases in higher-order association areas, we compute the spatial correlation of specificity with maps of cortical hierarchy (Fig. 3F). In particular, gray matter myelination, measured by the T1w/T2w ratio, has been suggested as an indirect measure of the hierarchy from sensory to association areas[30,41]. We find a significant negative correlation of specificity with the T1w/T2w ratio (Fig. 3F) indicative of increased specificity along the sensory-to-association hierarchy[30,41]. This correlation holds when the analysis is conducted on a more fine-grained atlas and when testing significance with permutation statistics accounting for the spatial autocorrelation

of cortical myelination across adjacent brain areas (Fig. S13). In addition, specificity is negatively correlated with the second principal gradient of functional connectivity, which separates somatomotor and auditory cortices (Fig. S13C)[42]. There was no consistent association of specificity to the first principal gradient of functional connectivity capturing sensorimotor-to-transmodal progression (Fig. S13B)[42].

## Novelty across saccades

As with film cuts, BHA responses associated with saccades are widespread across the brain, with long latencies suggestive of semantic visual processing (Figs. 2 and S3). We hypothesized that responses associated with saccades are modulated by changes in low-level and semantic visual features between the foveal image before and after each saccade. If true, we would expect stronger saccade-related responses when the image features before and after a saccade are less similar to one another, i.e. when the upcoming target of a saccade is novel. To measure novelty we leverage a deep convolutional neural network trained with contrastive learning. Specifically, we use a ResNet pre-trained to extract features shared across different image patches[29]. These features capture semantic properties of objects in the images[43]. We compute the euclidean distance of features for image patches of 5×5 degree in visual angle around the gaze position before and after each saccade (Fig. S14A, B). In doing so, each saccade is associated with a numerical value indicating the novelty of the upcoming fixation. Interestingly, on average the novelty associated with observed saccades was larger than random simulated saccades, matched in distance and direction (Fig. S14C). This indicates that viewers tend to direct their gaze towards locations with higher novelty. This is particularly true shortly after film cuts, whereas saccades later during scenes tend to move towards low-novelty targets (Fig. S15).

Next, we divided all 55,334 saccades from all 23 patients and videos into two equally sized groups with high and low novelty (Fig. 4C), while controlling for saccade amplitude (Fig. S14D). We excluded saccades across cuts and the first saccade after each cut. We predicted that saccades with high novelty will be related to stronger BHA responses. We estimate the magnitude of the response to each saccade as before (Fig. S8). We find significant main effects for saccade novelty and brain region in a mixed-design ANOVA with patients as a random factor (Fig. 4A, $F_{(1,1632)} = 19.77$, $p = 9.3*10^{-6}$; Region: $F_{(5,1632)} = 20.41$, $p = 8.2^{-20}$, Table S7). A follow-up mixed-design ANOVA for each brain region shows that novelty increases response amplitude in occipital and temporal lobes (Fig. 4A, occipital: $F_{(1,264)} = 9.06$, $p = 0.0043$, temporal $F_{(1,513)} = 11.83$, $p = 0.0011$, FDR corrected at q = 0.05, Table S8). An analysis of the amplitude difference in smaller parcels shows that most of the cortex responds stronger to high-novelty saccades. However, there are some exceptions, particularly in the parietal and temporal lobes, responding stronger to low-novelty saccades (Fig. 4B). We also computed separate TRFs for saccades with high and low novelty, including film cuts and optical flow as regressors to remove correlated activity (Fig. 4C). In the occipital lobe the majority of channels are associated with responses to either high or low novelty, i.e. respond with low specificity (Fig. 4D). In contrast, in the frontal lobe most channels are responsive to either high or low novelty saccades (Figs. 4D and S17), with a specificity above chance (median = 1, $p = 0.024$; 1000 permutations. FDR corrected across brain regions at q = 0.05, Table S9). This suggests a specialization for high novelty, but at the same time, low-novelty saccades. Interestingly, in the parietal and frontal lobes, some responses related to low-novelty saccades show an inhibition of neural activity (Fig. S16). In contrast, channels responsive to high-novelty saccades show only increases in neural activity. The comparison of specificity with the T1w/T2w ratio leads to strong negative correlations indicating that specificity increases along the sensory-to-association hierarchy (Figs. 4E, F and S18A). In addition, specificity of responses to high- and low-novelty

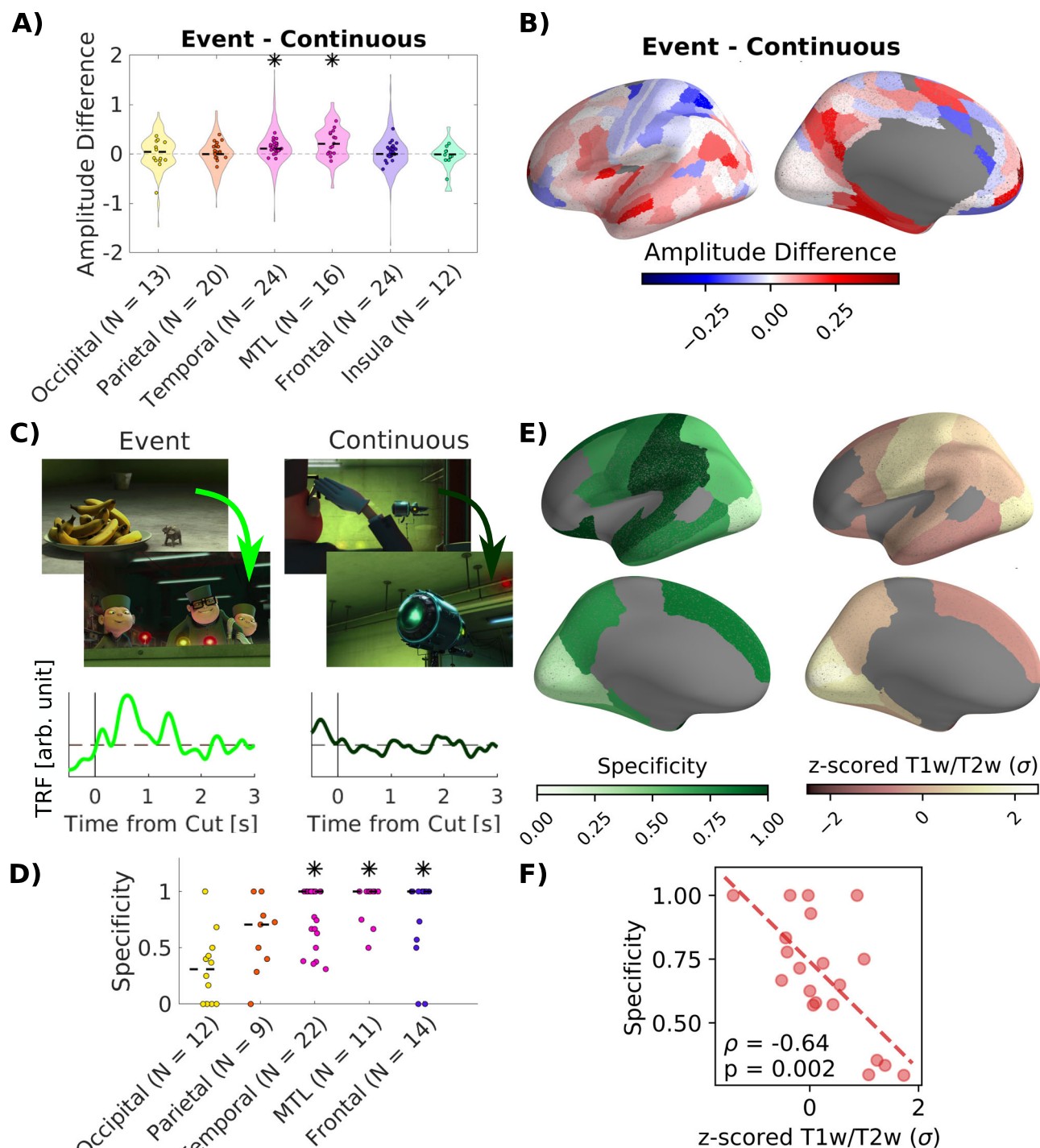

**Fig. 3 | Neural responses associated with film cuts are stronger, and in higher-order association areas respond specifically to event cuts as opposed to continuous cuts. A** Difference in magnitude of the responses to individual event cuts and continuous cuts (Fig. S7). Magnitude of the response is estimated for individual events (Fig. S8). Only channels with significant responses to film cuts in Fig. 2 are considered. Violin plots show the distribution of the difference in magnitude of responses across channels, dots the average difference of magnitude within each patient. Stars indicate significant main effects of the type of film cut within regions with patients as a random factor in a mixed-design ANOVA (FDR corrected at q = 0.05, Table S5). **B** Weighted average of the amplitude difference of responses to event versus continuous cuts across significant channels within parcels of the Glasser atlas[35]. **C** Temporal response functions are obtained for two separate regressors coding separately for event cuts and continuous cuts. In this example channel (supramarginal gyrus) there is a significant response after 0.5 s to event

cuts, but no response to continuous cuts. TRFs in all channels are shown in Fig. S11. "Courtesy of Universal Studios Licensing. © 2010 Universal Animation Studios LLC and Universal City Studios LLLP. All Rights Reserved." **D** Specificity of responses in each region measured as a fraction of responsive channels responding to only one type of film cut. Dots show the specificity in each region for each patient. Stars indicate specificity that is significantly higher than expected by random assignment of channels to event or continuous cuts. Regions with only one significant channel have been removed from analysis. **E** Left: Specificity computed in parcels of the Desikan-Killiany atlas[89]. Specificity is computed in all parcels with at least five significant channels. Right: Mean T1w/T2w ratio in each parcel of the Desikan-Killiany atlas. **F** Spearman correlation coefficient between T1w/T2w ratio and specificity (p-value for two-tailed test). Results in a finer grained parcellation, as well as correlations with gradients of functional connectivity are shown in Fig. S13. Source data are provided as a Source data file.

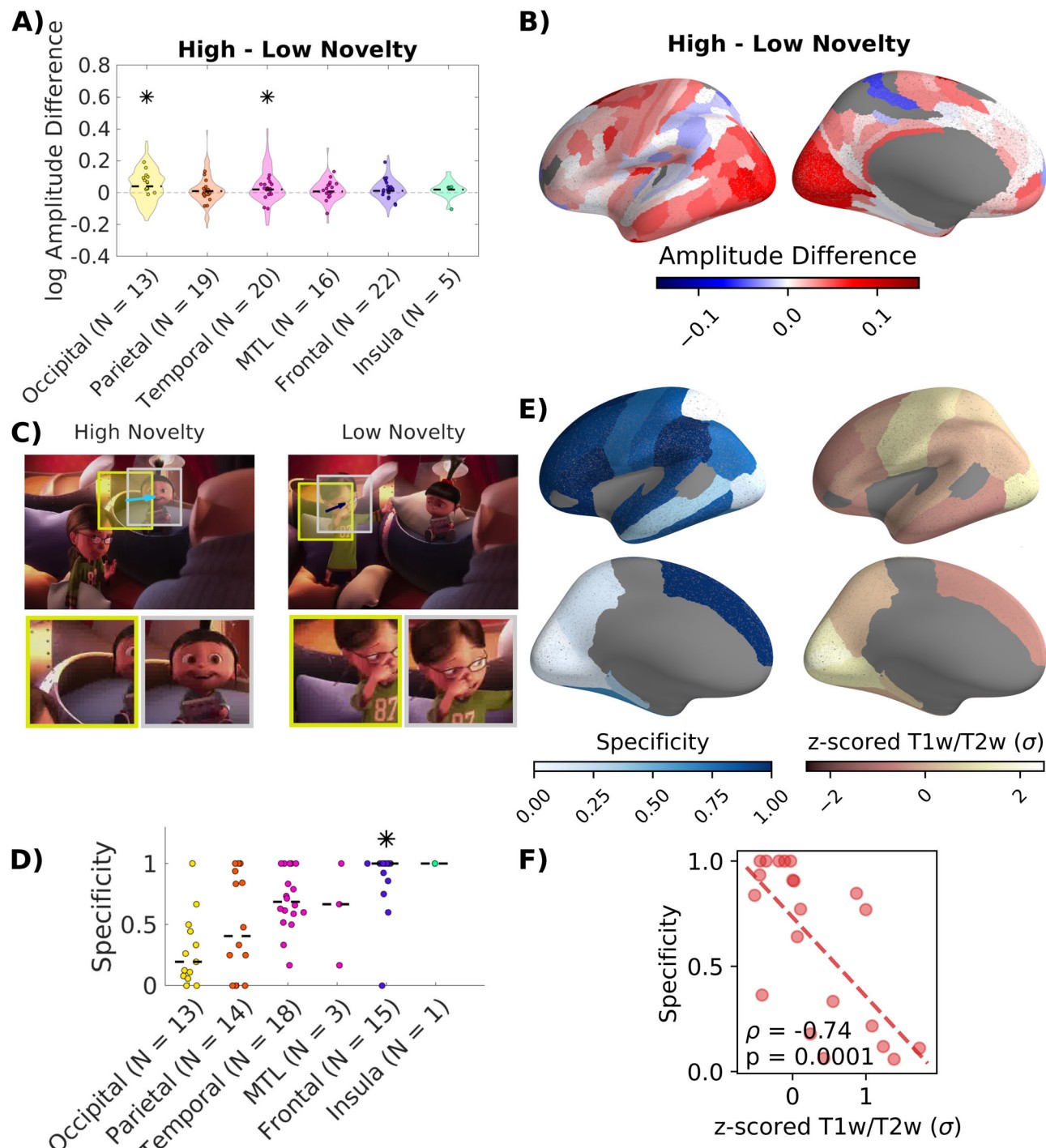

**Fig. 4 | Differential responses associated with saccades to high and low novelty visual stimuli in higher-order association areas. A** Difference in magnitude of neural responses to individual saccades with high and low novelty targets. Magnitude was estimated using filters for all saccades as described in Fig. 3A. Stars indicate significant main effects of the type of film cut within regions with patients as a random factor in a mixed-design ANOVA (FDR corrected at q = 0.05, Table S8). **B** Weighted average of the amplitude difference in parcels of the Glasser atlas[35]. **C** Temporal response functions are estimated for saccades with high and low novelty separately. Example of a high-novelty saccade (feature distance = 8.49) and a low-novelty saccade (feature distance = 4.85) with similar saccade amplitude. Novelty is computed as the distance between features from a convolutional neural network (Fig. S14). TRFs in all channels are shown in Fig. S16. "Courtesy of Universal Studios Licensing. © 2010 Universal Animation Studios LLC and Universal City Studios LLLP. All Rights Reserved." **D** Specificity of responses to saccades with high and low novelty defined as in Fig. 3D. **E** Left: Specificity computed for channels in each parcel of the Desikan-Killiany atlas[89]. Parcels with less than five responsive channels are not included. Right: Map of T1w/T2w ratio, a proxy of gray matter myelination[30,41]. **F** Spearman correlation of specificity and the T1w/T2w gradient (p-value for two-tailed test). Spatial correlations of specificity with the T1w/T2w map in a finer grained parcellation, as well as correlations with gradients of functional connectivity are shown in Fig. S18. Source data are provided as a Source data file.

saccades is related to the second gradient of functional connectivity, separating somatomotor and auditory cortices (Fig. S18C). There was no consistent association with the first gradient of functional connectivity (Fig. S18B). As such, these results parallel those for specificity to the type of film cut.

Behavioral data shows that saccade novelty does not significantly differ between repeats of "The Present" when controlling for saccade amplitude (difference of median novelty: $-7*10^{-4}$, p = 0.46, $N_{repeat\_1}$ = 6,843, $N_{repeat\_2}$ = 5,121, Wilcoxon rank sum test). However, saccade amplitude increases significantly in the second repeat of "The Present" (difference of median amplitude 0.55 DVA, p = $1.5*10^{-31}$, $N_{repeat\_1}$ = 6,843, $N_{repeat\_2}$ = 5,121, Wilcoxon rank sum test).

### Saccades to faces

A more direct way to quantify semantic changes across saccades is by the content of the saccade target. Specifically, saccades to faces have been reported to relate to stronger neural responses than saccades to other objects[10]. To detect faces in the movies we finetuned a pretrained object detection and segmentation network with a subset of labeled frames from our videos. This network was then used to detect faces in the unlabeled frames. We then divide saccades to faces and saccades to other objects (non-face saccades, 10,225 each) and analyze their neural response functions (Fig. S19).

As above we calculate the difference in the magnitude of responses related to face and non-face saccades. We find a significant main effect of face vs non-face and brain region in a mixed-design ANOVA with patient as random effect (Face vs non-face: F(1,1551) = 40.4, p = $2.72*10^{-10}$; Brain Region: F(5,1551) = 11.46, p = $6.99^{-11}$, Table S10). Surprisingly, the responses related to non-face saccades have a significantly larger magnitude than those related to face-saccades in the occipital, parietal, mediotemporal (MTL) and frontal lobe (Fig. 5A) (follow-up mixed-design ANOVA for each brain region with patient as random effect, Fig. 5A: Occipital: F(1,263) = 30.84, p = $1.65*10^{-7}$; Parietal: F(1,200) = 19.01, p = $3.57*10^{-5}$; MTL: F(1,100) = 9.26, p = 0.0036; Frontal: F(1,432) = 13.43, p = $3.72*10^{-4}$; FDR control, at a level of q = 0.05, Table S11). This is surprising because saccades to faces have a higher novelty (p = $1.6*10^{-60}$, $N_{face}$ = 7636, $N_{non-face}$ = 7784, Mann-Whitney U-test), so if anything, the opposite effect would have been expected. Consistent with this, computing the weighted average of the amplitude difference of responses confirms stronger responses to non-face saccades in most brain areas (Fig. 5B). However, this fine grained analysis also shows some areas in the parietal and temporal lobes with larger responses to face saccades (Fig. 5B) To determine the specificity of responses we compute separate TRFs for face and non-face saccades, including film cuts as a regressor to remove correlated activity. Most channels throughout the parietal, temporal, frontal lobe and insula are selectively responsive to either face or non-face saccades (Fig. 5D, Parietal: median = 0.94, p = 0.002; Temporal: median = 0.97, p = 0.002; Frontal: median = 1, p = 0.002; Insula: median = 1, p = 0.025. 1000 permutations. FDR correction at q = 0.05, Table S12). Interestingly, in the superior temporal gyrus, which contains the auditory cortex, the largest fraction of responsive channels is to face-saccades (Fig. S20). The spatial correlation of specificity with the T1w/T2w map is not significant for larger parcels in the Desikan-Killiany atlas (Fig. 5F). This result is unsurprising given that responses across the brain are specific to non-face saccades starting from sensory areas (Fig. 5B, D). There is a significant correlation, however, of the specificity to face and non-face saccades and the second functional gradient, which separates somatomotor and auditory cortices (Fig. S21). Overall, these results indicate that faces and other objects are associated with selective responses in brain areas extending far beyond traditional visual processing areas. However, again, we can not rule out that some of these responses are due to other unobserved factors associated with the presence or absence of faces.

### Face motion

Responses to motion are dominated by responses related to face motion compared to total motion (Fig. S22). Specifically, face motion is associated with responses in distinct clusters of channels in the lateral occipital cortex, fusiform gyrus and superior temporal sulcus (Figs. S22 and S23). These areas include face-processing areas[44–46], known to respond stronger to moving faces[4,26,44]. Other areas, however, show specific responses related to total motion, notably the Hippocampus (Fig. S23), which might reflect spatial remapping during camera movements[47].

## Discussion

Much of the existing neuroscience literature related to motion perception, saccades or dynamic visual stimuli focuses on the effects of individual stimulus properties on neural activity in anatomically constrained brain areas. This approach links specific effects to existing reports for the same brain areas. However, implicitly, it also ascribes a narrow specialization to individual brain areas which may not be warranted in a real-life setting. Here we take a more inclusive approach to study the effect of multiple sources of visual change in a more ecologically valid setting of watching movies. The system identification approach controls for the observed correlation between motion, saccades, and film cuts. We found that low-level and semantic visual changes across film cuts and saccades result in fluctuations of neural activity in distinct and widespread neural populations across the whole brain. In contrast, motion appears to affect a more confined area of visual brain regions. In the following we will put these results in context of the specific literature in each of these areas.

Overall, responses related to motion are more confined to visual areas compared to responses related to film cuts and saccades (Figs. 2D and S4). Therefore, motion in naturalistic stimuli is perhaps a less dominant driver of neural responses than previously thought[3,4]. The responses related to motion we find in our data, however, are consistent with previous reports. Studies using simple motion stimuli as well as naturalistic stimuli have identified specific motion sensitive brain areas in the occipital, temporal and parietal lobes, such as the medial temporal area (MT), or the ventral intraparietal area (VIP)[3,4,39,48]. However, motion in naturalistic stimuli activates much wider areas across the temporal lobe, largely driven by motion of socially relevant stimuli, such as faces or body parts[3,4,38]. Similarly, we find widespread responses associated with optical flow in occipital, temporal and parietal lobes (Fig. 2).

We find that responses associated with film cuts, especially those at event boundaries, are widespread across the entire brain (Figs. 2 and 3). These results consolidate previous findings of responses to event boundaries in individual brain areas, most prominently in the medial temporal lobe. This supports the proposed role of event boundaries in organizing memory of continuous experience[6,20,22,23,40,49]. Our results show that event cuts elicit stronger responses (than continuous cuts) in association areas like the angular gyrus and posterior medial cortex (Fig. 3B), consistent with previous fMRI results[22]. We find the opposite effect in the superior parietal and premotor cortex (Fig. 3B), similar to a previous fMRI study[21]. This is consistent with the suggestion that the superior parietal cortex maintains internal sensory and motor representations[50]. While film cuts are not usually considered in the context of event boundaries[5,51], recent work has identified distinct responses depending on the nature of visual change across film cuts. For example, different neural populations in the medial temporal lobe show specific responses to film cuts within or between different video clips[6]. In addition, film cuts with action discontinuities activate parietal and frontal areas[21], while film cuts in general strongly activate visual areas[5,21]. Here we propose that while all film cuts are associated with various changes in low-level visual content, some coincide with semantic change at event boundaries. We defined the latter as "event cuts" and cuts within events as "continuous cuts" based on standard

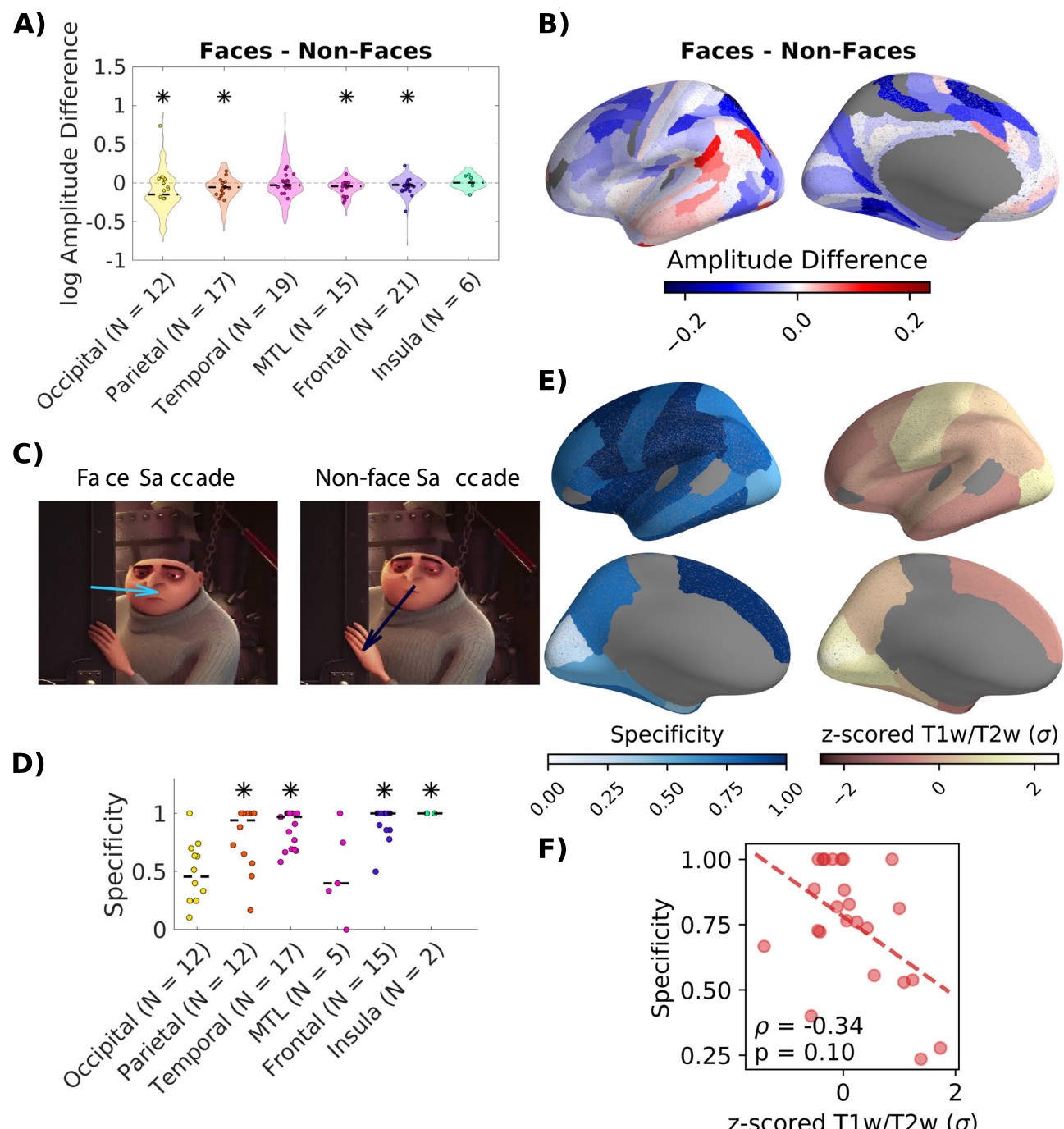

**Fig. 5 | Responses that are specific for face and non-face saccades are found in higher-order association areas. A** Difference in magnitude of neural responses to individual saccades to faces and non-faces. Magnitude was estimated using filters for all saccades as described in Fig. 3A. Stars indicate significant main effects of the type of film cut within regions with patients as a random factor in a mixed-design ANOVA (FDR corrected at q = 0.05, Table S11). **B** Weighted average of the amplitude difference in parcels of the Glasser atlas. **C** Separate TRFs are computed for face and non-face saccades (Fig. S19). "Courtesy of Universal Studios Licensing. © 2010 Universal Animation Studios LLC and Universal City Studios LLLP. All Rights

Reserved." **D** Specificity of responses to saccades to faces and non-faces defined as in Fig. 3D. **E** Left: Specificity computed for channels in each parcel of the Desikan-Killiany atlas[89]. Parcels with less than five responsive channels are not included. Right: Map of T1w/T2w ratio, a proxy of gray matter myelination[30,41]. **F** Spearman correlation of specificity and the T1w/T2w gradient (p-value for two-tailed test). Spatial correlations of specificity with the T1w/T2w map in a finer grained parcellation, as well as correlations with gradients of functional connectivity are shown in Fig. S21. Source data are provided as a Source data file.

event segmentation[20,21]. Our data shows that film cuts are associated with widespread neural responses across the whole brain (Fig. 2). Importantly, channels in the MTL and temporal lobe are related to stronger responses to event cuts than to continuous cuts (Fig. 3A). By showing that event boundaries can be linked to film cuts we were able to investigate their fast neural correlates. We show that event

boundaries are processed in a widespread ensemble with increased specificity in higher-order association areas (Fig. 3D).

Saccades create shifts of the image on the retina that result in neural responses in early visual areas[7,52,53]. These low-level responses are suppressed in higher-order visual areas, probably due to integration of visual responses with a copy of the motor signal[7,52,53]. In

contrast, previous work has found modulation of neural activity by saccades in the non-visual thalamus[9], medial temporal lobe[14,15], auditory, frontal and parietal cortices[16]. These findings collectively suggest that saccades are linked to the neural processing of higher-order cognitive processes, such as attention and memory[11]. Our findings support the view that neural activity across most, if not all the brain (Fig. 2), is modulated by saccades and in particular by their association with semantic novelty.

There is competing evidence that saccades in static visual scenes sample either semantically similar[54], or dissimilar objects in static natural scenes[55]. In movies, scenes change rapidly and we find that both types of saccades can be observed (Fig. S15). In static natural scenes, eye movements target objects with similar semantics[54]. We propose that this type of saccades corresponds to saccades with low semantic novelty in our video data. Saccades are also attracted by visual features that are novel in the context of static scenes[55]. We propose that this corresponds to saccades with high semantic novelty in our data. Exploitation of a visual scene by sampling similar objects competes with the necessity to explore novel objects in dynamic environments. We suggest that these objectives are concurrent and the balance may vary with task demands. For instance, we observed that saccade amplitude, a measure of exploratory behavior, increases during the second repeat of the video "The Present". In movies, eye movements are synchronized between subjects, suggesting a stereotypical behavior[56,57]. Increased saccade amplitude upon repeat could therefore indicate idiosyncratic exploration[57]. Interestingly the opposite behavior is found for static images which are less synchronized across subjects[56] and results in shorter saccades upon repetition[58]. Together this suggests that the balance between exploitation and exploitation depends on the task. Indeed, we find that after film cuts, saccades are directed primarily to novel targets (Fig. S15). We suggest that neural responses to high-novelty saccades support exploration, and responses to low-novelty saccades support exploitation. The concurrent demands are reflected in different sets of channels that are responsive to saccades with high and low semantic novelty (Fig. 4). The suppression of neural processing in the parietal and frontal lobes that we find in particular for low-novelty saccades (Fig. S16), may facilitate perceptual stability across eye movements[59].

Defining saccade novelty by face or non-face targets provides a clear link to semantic and social information related to faces. Faces contain a wide variety of features that are processed in distinct specialized areas[44]. Low-level, view specific features are processed in the occipital lobe, while face specific areas in the fusiform gyrus and superior temporal gyrus process face identity, face motion and eye gaze. An extended face network, distributed in the auditory cortex, limbic system and prefrontal cortex, has been proposed to process higher level semantic information, such as speech, directed attention, emotion and biographical information[10,16,60,61]. In support of this model, we find that saccades to faces and other objects are associated with activity fluctuations in distinct sets of channels across the whole brain (Fig. 5). In particular, we find responses related to saccades to faces in auditory areas, such as the STS, and the frontal cortex, as well as, distinct sets of channels responsive to face and non-face saccades in the medial temporal lobe (Fig. S20). Together these results show that the extended face network can be studied in real world scenarios, where multiple semantic features interact, by locking face processing to saccades.

In most recording locations we found responses to only one of the main sources of visual change (Fig. 2C). However, some locations respond to both film cuts and saccades, which may be due to unobserved common factors. For instance, after a film cut often a new face appears, but faces also attract saccades. Therefore, responses to faces may activate the same channels even after accounting for the correlation between film cuts and saccades. More generally, we note that in film there could be a variety of unobserved factors that are associated

with cuts, saccades or movement. For instance, stimulus movement is associated with sound. This could explain responses to motion we observed in the auditory cortex, including the transverse temporal (Heschl's) and superior temporal gyrus (Figs. S3 and S23)[62]. Therefore, we caution against the interpretation that these widespread responses are specific to cuts, saccades or motion. We also performed a number of contrasts individually for different types of film cuts, types of saccades and types of stimulus motion. These contrasts aim to capture the effect of semantic novelty, while keeping other factors constant (e.g. saccade amplitude). Nevertheless, there is a complex interaction between different features of the stimulus, as well as with saccade behavior so that confounds cannot be ruled out.

The relatively weak responses related to optical flow in our data could be due to several factors. First, our total motion feature is unspecific, capturing different types of stimulus motion. The face motion regressor captures only a small part of socially relevant motion. Thus, the related responses could be restricted in space. Second, electrode coverage in our patient population is not chosen to specifically cover motion sensitive areas. We further did not run any motion localizers. The locations we recorded from might not cover motion sensitive areas. However, we would still have expected widespread responses to motion. It is therefore possible that responses to film cuts and saccades simply are related to stronger neural responses than motion because they are associated with more pronounced changes and novel information.

The system-identification approach used here captures time-delayed responses only to first order. It does not capture higher-order (non-linear) effects on BHA or multiplicative interactions between regressors (motion, saccades and cuts). Techniques for doing this within a system-identification approach are readily available[33,34], but would lead to significantly more complex exposition. The current approach does, however, account and compensate for correlation in the regressors, contrary to more conventional reverse-correlation (evoked response) analysis.

We have focused on BHA as the best available correlate of neuronal firing that is available in iEEG recordings. One could perform a more thorough analysis of local field potentials, for example modulation of power in other frequency bands and phase alignment, which are a rich source of information on neuronal dynamics[63,64]. As shown by recordings in macaque V1, such analysis may be particularly important when analyzing modulatory top-down effects[8].

The analysis focused on saccade onset, rather than fixation onset. Our motivation was to look for change across a saccade, rather than focusing on the content of individual fixations. However, early on we determined that our analysis approach gives similar results if we used fixation onset instead of saccade onset (as the two are tightly coupled in time, $32 \pm 14$ ms, Fig. S6). However, we note that previous work showed that locking visual responses to saccade versus fixation onset highlight either top-down or bottom-up influences at the level of V1[8].

While we control for various low level features between event and continuous cuts, similar features could drive differences between responses to high- and low-novelty saccades. However, we control for saccade amplitude, which we expect to have the strongest influence on neural responses[65].

Including two repeats of "The Present" video in the same film cut TRF model might lead to temporally smeared responses, due to anticipatory responses[66] in the second repeat. While we present comics in Hungarian and English we have not investigated the effect of different languages. The narrative in comics is easy to follow regardless of language.

The human subjects are patients with epilepsy. We minimized the possible effects of this disease on our results by excluding epochs with epileptic activity and electrodes located in the seizure network from analysis and recording experimental data long after seizures when the electrophysiological signal and the patient's clinical status was back to

baseline. Furthermore, our patients have focal epilepsy and only ~18% of the implanted electrodes end up in epileptic tissue[28].

In summary, taking a data-driven approach to analyze intracranial EEG data during movies we found widespread responses related to film cuts and saccades. We show that semantic changes across film cuts and saccades can be defined through event boundaries, visual novelty and the presence of faces. These semantic changes modulate neural activity in distributed locations across the whole brain, particularly in higher-order association areas. This extends previously known anatomical locations with functional specificity to these visual features. Further studies will be needed to fully characterize the factors underlying these widespread yet specific responses related to visual change we have found here. The richness of film makes this analysis challenging, but we do believe that this richness is necessary to capture the full extent of neural processing of natural stimuli.

## Methods

### Dataset

Intracranial electroencephalography (iEEG) along with eye movements were recorded from 23 patients (mean age 37.96 years, age range 19-58 years, 11 female; Table S1) with pharmacoresistant focal epilepsy at North Shore University Hospital (Manhasset, New York). Patients were chronically implanted with depth and/or grid electrodes to identify epileptogenic foci. Three patients were implanted twice at different times. We recorded the same session twice with these patients with different electrode coverage. The study was approved by the institutional review board at the Feinstein Institute for Medical Research and all patients gave written informed consent before implantation of electrodes. Across patients electrode locations cover most of the brain (Fig. 1A). However, most dense coverage is available on the temporal lobe and coverage of the occipital lobe is more sparse (Fig. 1A). iEEG data was recorded continuously at 3 kHz (16-bit precision, range ± 8 mV, DC) on a Tucker-Davis Technologies data processor (TDT, Alachua, FL, USA). Gaze position was recorded simultaneously with iEEG data with a Tobii TX300 eye tracker (Tobii Technology, Stockholm, Sweden) at 300 Hz. The eye tracker was calibrated before each video to prevent drift. We used parallel port triggers sent from the stimulus PC to the eye tracker and data processor to align the different data streams. A custom script for movie presentation in psychotoolbox (version PTB_Beta-2014-10-19_V3.0.12; Gstreamer version 1.10.2)[67], and the Tobii SDK for collecting eye tracking data were implemented in MATLAB (2012b on Windows 7). For additional accuracy in the alignment of the movie features to iEEG and eye tracking data we recorded timestamps at the onset of each frame with the clock of the eye tracker.

Patients watched up to 43.6 minutes of video clips (Fig. 1B). Video clips included segments of an animated feature film ('Despicable Me', two different clips, 10 min each, in English and Hungarian language), a animated short film with a mostly-visual narrative shown twice ('The Present, 4.3' min), and three clips of documentaries of macaques ('Monkey', 5 min each, without sound).

### Electrode localization

Each electrode shank/grid contains multiple recording contacts. Contact locations were identified using the iELVis MATLAB toolbox[68]. All subjects received a preoperative T1-weighted 1 mm isometric scan on a 3 T scanner. Tissue segmentation and reconstruction of the pial surface was performed with the freesurfer package[69,70]. Postoperative CT scans were acquired and coregistered to the freesurfer reconstruction. Contacts were then semi-manually localized using the bioimagesuite software (version 3.01)[71]. All contacts were then coregistered to the freesurfer fsaverage brain for visualization and assignment to anatomical atlases[72]. Subdural contacts were shifted to the closest vertex of the lepto-meningeal surface to correct for brain

shift while preserving the geometry of grid contacts. Freesurfer coordinates of subdural contacts are determined by finding the nearest vertex on freesurfer spherical pial surface. In contrast, stereotactic electrode shanks were coregistered to fsaverage space using a linear affine transformation. Stereotactic contacts close to the pial surface (<4 mm) are assigned to cortical atlases by finding the nearest vertex on freesurfer spherical pial surface. For further analyses stereotactic contacts close to the pial surface were shifted to the nearest vertex of the native pial surface and then moved to fsaverage space in the same manner as subdural contacts.

### Data preprocessing

iEEG data was minimally processed by removing line noise at 60 Hz, 120 Hz, and 180 Hz, with a 5th order butterworth bandstop filter, and low frequency drift at 0.5 Hz with 5th order butterworth high-pass filter. The data was then re-referenced to a bipolar montage. Artifacts with an absolute voltage 5 times of the interquartile range of voltage of each channel were removed. Further, after visual inspection, channels with spiking activity and channels outside the skull were identified manually and removed from analysis. The power of the signal in each frequency band is calculated by the absolute value of the Hilbert transformation of the bandpass filtered signal. The broadband high-frequency amplitude (BHA) power is defined in the range of 70-150 Hz. The power envelope is then downsampled to 60 Hz.

### Stimulus motion

As a measure of motion we extract optical flow from each video using the Horn-Schunck method as implemented in MATLAB[73]. The Horn-Schunck method computes the displacement vectors of pixels from one frame to the next, assuming smooth flow across neighboring pixels. We average the displacement vectors across all pixels within each frame. This results in a regressor of average motion throughout the video (Fig. 1C). The motion regressor contains artifacts from film cuts. To remove these artifacts, we replace the samples within a window of 166 ms around film cuts with a linear interpolation. For movies with sound ("Despicable Me English", "Despicable Me Hungarian", "The Present") we correlate the motion regressor with the sound envelope. The sound envelope is calculated as the absolute value of the Hilbert transform of the sound extracted from the movie files. The correlation is computed across the concatenated signal for all three movies. To test the significance of this correlation we compute a surrogate distribution of the motion regressor by 10,000 circular shuffles.

### Film cuts

Film cuts in the movies were identified as peaks in the temporal contrast between consecutive video frames. Temporal contrast is the mean square difference of luminance between consecutive video frames[3]. Film cuts with smooth transitions do not show up as sharp peaks in the temporal contrast and are missed. Moments of sudden motion on the other hand, might be mistaken for Film cuts. Film cuts detection is corrected by visual inspection to account for these errors.

### Event cuts versus continuous cuts

To classify film cuts based on changes in semantic information we align film cuts to event boundaries. We record event boundary annotations for all videos in a separate study conducted online. 200 participants were recruited on Prolific (www.prolific.co). The task was implemented in PsychoJS scripts created from the psychopy builder (Psychopy3, version 2021.1.4)[74]. The task was hosted on Pavlovia (https://pavlovia.org/). Participants watched one of the videos each with the following instructions: "The movie can be divided into meaningful segments. You will have to indicate when you feel like a meaningful segment has

ended by pressing the 'spacebar. You will likely detect multiple events throughout the movie.". For all videos, except Despicable Me, we included a task to check attention. Participants saw a black screen with 10 white boxes flashed at random times. Participants had to respond with a button press every time they saw a white box. We excluded the data from 20 out of 200 participants because either no event boundaries were annotated or the attention test failed. The attention test was considered failed if participants responded to less than 8 of the white boxes in the task after the movies.

Event boundary annotations are delayed after film cuts due to reaction time and processing of visual information. To adjust for this, 1 s was subtracted from every event boundary (similar to[23]). Event boundaries from all participants were aggregated in one vector consisting of impulses at the time of button presses per video. This vector was then smoothed with a Gaussian of 0.5 s, to account for variance in the manual annotation of event boundaries (Fig. S7A)[23,24]. The resulting vector is a measure of event salience (Fig. S7A)[24]. Event salience in our data is consistent with salience from data collected by Cohen et al. (Fig. S7B, C)[40]. This allowed us to compute event salience at the time of each film cut. Film cuts were sorted by event salience. 'Event cuts' are the film cuts with event salience above a threshold. For each movie this threshold is defined as the change point detected with the findchangepts() function in MATLAB. This method minimizes the residual error from the mean in the segments before and after the change point. We select 57 event cuts out of a total of 561 cuts (Table S3). An equal number of cuts with lowest salience were selected, which we refer to as 'Continuous cuts'. We test if event cuts and continuous cuts differ in changes of low-level visual features[6]. These features are luminance, contrast, complexity, entropy, color, and features from layer fc7 of AlexNet[6]. Complexity was quantified as the ratio of file size after JPEG compression[75]. We compute a p-value for each feature across cuts with a Wilcoxon signed rank test and correct for multiple comparisons with the Benjamini-Hochberg procedure. If event and continuous cuts differ in low-level features we iteratively select a random set of continuous cuts from the film cuts in the lower half of event salience until there is no difference in low-level features.

## Face detection
We use an object detection algorithm made available through facebook's Detectron2 platform[76]. We selected a ResNeXt-101-32x8d model backbone[77] trained in the mask R-CNN framework[78] on the COCO dataset[79] due to its high segmentation accuracy compared to other models on the Detectron2 platform v0.5. Neural networks for face detection exhibit high performance on natural movies, however, face detection in comics requires retraining of the networks. We therefore annotated faces in 4551 frames in 'Despicable Me English' and 1575 frames in 'Despicable Me Hungarian' using the 'Labelme' version 4.5.6 and 'Roboflow' (Roboflow Inc, Des Moines, Iowa) tools. We applied flip and 90° rotations for data augmentation and created a training and validation set with a 80%-20% train-validation split ratio. For 'Despicable Me English' achieved a mean average precision of bounding box annotations mAP=0.61 and classification accuracy 74.5% on the validation dataset (mAP=0.74 and 80% classification accuracy on a subset of frames). For 'Despicable Me Hungarian' we achieved a mAP=0.58 and a classification accuracy of 78% on the validation dataset. Missing bounding boxes and wrong labels were corrected manually. Faces in the video 'The Present' were annotated manually with 'Labelme' in the whole video.

## Saccade detection
For saccade detection we apply a 20th order median filter to smooth the gaze position data and compute eye movement velocity. Samples of the eye velocity that were faster than 2 standard deviations from average eye velocity were labeled as saccades. Often we observe a short adjustment of the eye movement after a saccade until it fixates on the new target. We correct this overshoot by merging these samples to the saccade. To combine samples for the saccade and the overshoot, we perform a morphological closing operation with a kernel size of 5 samples (16.7 ms) on the samples belonging to the saccade and overshoot. We label the first sample in the saccade as the saccade onset. The fixation onset corresponds to the first sample after which eye velocity drops under the 70th percentile, computed from velocity values within 33 ms before and 120 ms after saccade onset. The eye tracker provides labels for data quality when the gaze was not detected, for example during eye blinks. Saccades within 83 ms of samples with low data quality are removed. We also remove saccades within 110 ms after a previous saccade.

## Classification of saccades to faces
Saccades to faces could be determined simply using the location of the fixation onset. If the saccade lands on the bounding box of a face annotation the saccade could be classified as a face saccade. However, several saccades move towards faces but land just outside the face bounding box (Fig. S19A). Other saccades land within a face bounding box, but move away from the center of the face (Fig. S19B). Therefore, we generate several handcrafted features and classify face and non-face saccades using an SVM (Fig. S19). The first feature is a binary variable indicating whether the saccade points towards or away from the centroid of the closest face annotations bounding box. The second feature measures the overlap of a circle with a radius of 5 degree visual angle with all face annotation bounding boxes. The third feature is the distance to the closest face annotation centroid. The fourth feature is the angle between the vector from saccade onset to the face annotation centroid and the vector from saccade to fixation onset. The fifth feature is the angle between the vector from saccade onset to the face annotation centroid and the vector from fixation onset to the face annotation centroid. We manually label a total of 1288 face and non-face saccades for saccades in one video from one patient to obtain training data. We fit an SVM with a Gaussian kernel in MATLAB using fitcsvm() and a kernel scale of 2.2. We achieve a cross-validation accuracy of 0.964 using 10 fold cross-validation. Saccades in all other videos and patients are classified in face and non-face saccades using this SVM model. The SVM classifies saccades and provides a score, indicating the signed distance to the decision boundary. Negative scores indicate saccades predicted as non-face saccades. Saccades with scores above 1 are classified as face saccades.

## Saccade novelty
To quantify the change of semantic novelty across saccades we use convolutional neural networks trained through contrastive learning[29]. In contrastive learning neural networks are trained on subsets of transformation of images, in order to learn more generalizable representations. The networks are trained to minimize the distance of features from transformations of the same image. The most useful transformations to improve performance are random crops of images[29]. These random crops are similar to saccades in images. In fact, crops based on simulated saccades improve performance of networks trained with contrastive learning compared to random crops[80]. Here, we compute the feature distance between image patches extracted around gaze position at saccade and fixation onset (Fig. S14A). Patches have a size of 200×200 pixels corresponding to the size of the foveal visual field of 5 degree visual angle. Features of a convolutional neural network of the pre- and post-saccadic patch are computed with a ResNet-50 trained with SimCLR version 1[29]. Saccade novelty is then defined as the euclidean distance between the features of the pre- and post-saccadic image patch (Fig. S14B). A large distance between features corresponds to high saccade novelty. Novelty, thus defined, trivially correlates with saccade amplitude as patches further away will look less alike (Fig. S14D). Thus, we divide all saccades into two groups

of high and low saccade novelty, while controlling for saccade amplitude. We do this by fitting a linear regression model to describe the relationship between saccade novelty and saccade amplitude. Saccades with higher novelty than predicted with this linear model comprise the group of saccades with high novelty (Fig. S14D). The groups of saccades with high and low novelty are matched in number. To control for the possible confound of film cuts, saccades across film cuts and saccades within 1 seconds after film cuts are removed from the analysis.

### System identification approach to establish temporal response functions

We identify neural responses to features in the movies with a conventional linear system identification approach, implemented in the mTRF toolbox version 2.0[33,34]. Each channel is analyzed individually. The inputs to the system are the time courses for motion, film cuts, and saccade onset (Fig. 1C). These are the same for all channels (from a patient). The output is the time course of the BHA neural signal for every channel. All signals are (re)sampled at 60 Hz—twice the frame rate. An impulse response, or temporal response function (TRF), is estimated (with ordinary leasts squares with ridge regression with 0.3 as the ridge parameter: Supplementary Eqs. (1) and (2), ref. 33) that maps the stimulus to the BHA signal through a convolution (Fig. S2A). For each channel TRFs are estimated simultaneously for all inputs to remove correlation. We fit the TRFs in with latencies from 0.5 seconds before to 3 seconds after the visual stimulus. After estimation, TRFs are smoothed with a Gaussian window with a standard deviation of 53 ms to filter higher-frequency noise. Similar smoothing is common in cluster-based statistical analysis often performed in fMRI research and increases sensitivity of the analysis[81,82]. To determine responses to semantic change associated with film cuts, saccades and motion we construct TRF models with different regressors. Different types of film cuts were modeled as event cuts and continuous cuts as inputs, also including film cuts to control for correlated activity (Fig. 3). We found that including motion as an additional regressor did not change results (Fig. S9). The model to test differences of novelty across saccades includes separate inputs for saccades with high and low novelty, as well as film cuts and motion (Fig. 4). To test differences in saccades by saccade target we construct a model with separate inputs for face and non-face saccades, also including film cuts (Fig. 5).

### Statistically significant responses

To determine the statistical significance of responses, i.e. predictable fluctuations in BHA, we compute a surrogate distribution of TRFs with time-shuffled output signals. Surrogates output signals are constructed by random circular shifts in time of the BHA in each electrode (i.e. multiple channels). Input signals are left unchanged to preserve their correlation structure. For all analyses we construct 10,000 surrogate outputs. Surrogate TRFs are computed as above using the surrogate output signals. We then determine which channels have time points in the TRF that are significantly different from surrogate TRFs. We refer to it as a significant response. This is done separately for each input (regressor). Thus, for a given channel, we may find a significant response for saccades (i.e. the saccade TRF has a significant time point) or we may find a significant response for cuts, or the responses may be significant for both (i.e. both TRFs have a significant time point). Corrections for multiple comparisons across time points and channels are addressed through cluster-based statistics[82]. Significant clusters are determined in two steps. First, we determine a test statistic α for each time point as the proportion of surrogate TRFs that have a more extreme amplitude than the original TRF. Clusters are defined as connected time points and channels on a shaft/grid that satisfy the test statistic of α <0.001. For each cluster a weight is computed as the squared amplitude of the TRF summed over the cluster. Second, a distribution of surrogate weights is found for each electrode by taking

the maximum weight in each electrode. For each cluster in the original data we compute p-values as the proportion of surrogate weights that is larger than the cluster weight. Finally, the p-values for all clusters in all electrodes and patients are corrected for multiple comparisons using the Benjamini-Hochberg procedure implemented in mafdr() in MATLAB[83] at a false discovery rate of q < 0.05. Clusters with corrected p-values above 0.05 are considered significant. For example, 1151 channels showed significant responses to film cuts (Figs. 2 and S3). At a FDR of 0.05, this means that on average, 58 channels may be false discoveries.

To determine the extent of responses to cuts, saccades and motion we compute the ratio of channels with significant responses in each brain area and patient (Fig. 2D). Within each brain area we test whether the differences between the ratio of responsive channels to different regressors are statistically significant across patients with a Wilcoxon rank sum test. We correct for multiple comparisons across all p-values with the Benjamini-Hochberg procedure implemented in mafdr() in MATLAB[83] at a false discovery rate of q < 0.05.

### Response amplitude

To estimate the amplitude of the BHA responses in each channel to individual film cuts (or saccades) we fit the TRFs to the neural data in the same window around the specific film cut (or saccade) (Fig. S8). The amplitude is estimated using ordinary least squares regression. The regression coefficient describes the factor the TRF is multiplied with to best fit the neural data.

To test whether response amplitudes differ between conditions, for example event and continuous cuts, we conduct a mixed-design ANOVA at the level of individual channels. We model the fixed effect of condition (e.g. event or continuous cuts), fixed effect of brain region and random effect of patients as independent variables. To test in which regions response amplitudes differ between conditions we perform a follow-up comparison in each brain region with a mixed-design ANOVA at the level of channels. Here, we model the fixed effect of condition with the patient as a random effect. We control p-values for the main effects of condition and patients for multiple comparisons using the Benjamini-Hochberg procedure at a false discovery rate of q < 0.05.

### Specificity of responses

To measure whether responses are specific to condition (e.g. high vs low novelty in saccade target) we define a measure of specificity. Assume $N_1$ channels respond to condition 1, $N_2$ channels respond to condition 2, and among all those $N_{12}$ are responsive to both. Then specificity is defined here as a the fraction of responsive channels that respond to only one of the two conditions:

$$Specificity = \frac{N_1 + N_2 - N_{12}}{N_1 + N_2} = 1 - \frac{N_{12}}{N_1 + N_2} \qquad (1)$$

The second expression makes clear that this definition is one minus the Jaccard index, which is a standard measure of overlap. A specificity of 1 indicates that there is no overlap in responsive channels, while a specificity of 0 indicates that any channel responding to one condition also responds to the other.

To test whether specificity differs between brain regions we apply a Skillings-Mack test[84], which is a non-parametric alternative to a two-way ANOVA for unbalanced data. This test determines statistical significance of the fixed effect of brain regions on specificity, while controlling for the random effect of patients.

If responsive channels were selected at random to respond to either condition, we may find by chance some level of specificity. To test whether the observed level of specificity in a given brain region is larger than this chance value we construct a surrogate distribution of specificity by shuffling the assignment of responsive channels to a

condition at random. For instance if 7 channels in a given brain region and patient respond to event cuts, 5 respond to continuous cuts and 2 to both, we create 1,000 permutations of 10 channels. In each permutation the number of channels responding to event cuts, continuous cuts or both is randomly assigned. This results in a surrogate distribution of median specificity in each brain area. We test the median across patients of the original data against this surrogate data to obtain a p-value. P-values are corrected for multiple comparisons across brain regions with the Benjamini-Hochberg procedure at a false discovery rate of q < 0.05. We remove regions with only 1 responsive channel in any given patient from analysis, because no distinct permutations can be constructed.

### Removal of saccadic spike artifacts
Significant TRFs to saccades often consist of sharp spikes at the time of the saccade (Fig. S24B). These channels are localized close to the orbit of the eyes and are likely artifacts of muscle movements Fig. S24C[85]. To remove these channels from all further analysis we construct a correlation matrix between all significant TRFs (Fig. S24A). We then cluster this correlation matrix to visually identify the group of channels with saccadic spikes (Fig. S24A, B)[86]. Saccade related artifacts are identified and removed from all analyses with this method.

### Registration of responses to cortical atlases
Our analysis is based on bipolar channel pairs. In the spatial plots all contacts that are part of a significant bipolar channel pair are included. In order to localize effects to specific anatomically or functionally defined brain areas we adapt the pipeline developed by Gao et al.[30] (https://doi.org/10.5281/zenodo.4362645). Electrode coordinates from the iElvis pipeline described above are transformed from MNI305 to MNI152 space with an affine transformation[87]. In Fig. 1A electrodes within 4 mm distance to a parcel of the Glasser atlas are counted. For maps of weighted average responses in Figs. 2B, 3B, 4B, 5B and S4A we follow the approach described in Gao et al.[30]. Briefly, responses of all electrodes close to a parcel are weighted by distance following a Gaussian with full-width-half-max of 4 mm. Parcels where the highest weighted count is below 0.5 are excluded from analysis (gray parcels in plots). The pipeline in Gao et al. has been implemented with the Glasser atlas[35]. When analyzing specificity, responsive electrodes are sparse. To address this we utilize the coarser Desikan-Killiany atlas (Figs. 3E, 4E, 5E, S13, S18, and S21). Voxel resolution T1w/T2w ratio data is available through the neuromaps toolbox[88]. We transform this data onto the freesurfer surface using the neuromaps toolbox. This enables us to average T1w/T2w data within parcels of the Desikan-Killiany atlas available through freesurfer[89]. We follow the same approach to obtain functional gradients in the Glasser and Desikan-Killiany parcellation (Figs. S13, S18, and S21).

### Visualization
For visualization of the time-course in Fig. 2A-top, each TRF is z-scored and baseline corrected individually for each channel and each regressor (also corresponding panel in S3, S11, S16). The baseline is computed between 2.1 s and 3 s after stimulus onset (film cuts, saccades or motion). Electrodes are grouped in clusters of similar waveforms using the CBIG repository v.0.8.2[86,90]. For Fig. 2A-bottom we average similar TRF waveforms in the same cluster. Here, the scale is the same across all regressors for better comparison. For Fig. 2B we compute the maximum amplitude across time (preserving the sign) and average over all channels within a parcel of the Glasser atlas[35]. Positive values (red) indicate a dominant enhancement of TRF in that area. Negative values (blue) indicate a dominant suppression. Zero values (white) indicate that enhancement and suppression are similarly strong.

All surface plots have been created with code adapted from Gao et al.[30] and anatomical data from the neuromaps toolbox[88].

### Correlation to gradients
Registering results to standardized brain maps allows correlation of our results to anatomical and functional gradients. For brain maps in the coarser Desikan-Killiany atlas we compute the Spearman correlation coefficient and the associated p-value assuming that these larger parcellations provide independent measures. For brain maps in the finer Glasser atlas we follow Gao et al. and compute significance using permutation statistics taking into consideration spatial autocorrelation[30].

### Reporting summary
Further information on research design is available in the Nature Portfolio Reporting Summary linked to this article.

## Data availability
The processed results data are available at OSF (https://doi.org/10.17605/osf.io/n6vpc). Source data are provided with this paper.

## Code availability
All data analysis was performed in Matlab 2020b and Python 3.10.9. Code for analysis is provided on github (https://doi.org/10.5281/zenodo.7811028) and is built on several packages: The mTRF toolbox version 2.0 (https://github.com/mickcrosse/mTRF-Toolbox/releases/tag/v2.0), Freesurfer version 6 (https://surfer.nmr.mgh.harvard.edu/fswiki/DownloadAndInstall), and the iELVis toolbox (https://github.com/iELVis/iELVis) for electrode location and visualization; iElvis uses Bioimage Suite version 3.01 (https://medicine.yale.edu/bioimaging/suite/lands/downloads/); The CBIG toolbox v.0.8.2 for clustering responses (https://github.com/ThomasYeoLab/CBIG/releases/tag/v0.8.2-Yeo2011_fcMRI_clustering); Labelme version 4.5.6 (https://github.com/wkentaro/labelme/releases/tag/v4.5.6) and Roboflow (Roboflow Inc, Des Moines, Iowa) for face annotation. Face detection was performed using the detectron2 platform v0.5 (https://github.com/facebookresearch/detectron2/releases/tag/v0.5). Saccade novelty was computed with a ResNet-50 trained with SimCLR version 1 (https://github.com/google-research/simclr/releases/tag/1.0). Cortical surface plots are based on field_echos v1 by Gao et al.[30] (https://doi.org/10.5281/zenodo.4362645). Maps for functional and anatomical gradients were obtained with neuromaps v0.0.3 (https://doi.org/10.5281/zenodo.7154329).

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

## Acknowledgements

This research was supported by the National Institutes of Health grants R01MH111439 (M.N., M.L., B.E.R., L.H., N.M., K.S., C.E.S., A.D.M., S.B., L.C.P.), P50MH109429 (M.N., M.L., B.E.R., L.H., N.M., K.S., C.E.S., A.D.M., S.B., L.C.P.), and R01CA247910 (L.H., K.S., L.C.P.), and National Science Foundation grant DRL-2201835 (L.H., K.S., L.C.P.). The authors thank Atanas Stankov and Jens Madsen for their technical and intellectual input concerning data analysis, interpretation and core scientific ideas. We thank Elizabeth Espinal, Sabina Gherman and Gelana Tostaeva for their assistance in data collection. Additionally, the authors thank Tejaswini Sudhakar, Mohigul Nasimova, Raydi Camilo Jimenez, Rasha Hussain, Nicole Zheng, Justin Lam and Samantha Lee for tirelessly annotating faces in all movies.

## Author contributions

Conceptualization: M.N., L.C.P., C.E.S.; methodology: M.N., M.L., B.E.R., L.C.P.; software: M.N., L.H., N.M., K.S.; formal analysis: M.N.; data curation: M.N., K.S.; data collection: M.N., M.L., S.B.; visualization: M.N., N.M.; supervision: L.C.P., C.E.S., A.M.; writing—original draft: M.N., L.C.P.; writing—review and editing: M.N., M.L., B.E.R., L.H., N.M., C.E.S., S.B., L.C.P.

## Competing interests

The authors declare no competing interests.
