## [Peer Review File · Nature Communications]

Semantic novelty modulates neural responses to visual change across the human brainREVIEWER COMMENTS

Reviewer #1 (Remarks to the Author):

Summary

This study by Maximilian et al used intracranial EEG data in humans watching some movie clips and analyzed whole brain responses based on high gamma activity. 1. estimate temporal response function (TRF) of stimulus motion, film cut, and saccade and TRF's distribution across brain regions; 2. divide the film cut into event cut and continuous cut, and reveal this event boundary segmentation response in the whole brain; 3. Finally, the author focuses on the saccade in detail. On the one hand, by comparing the semantic distance of the pictures before and after the saccade to capture the electrode distribution of the semantic novelty. On the other hand, the distribution of social information novelty was analyzed by comparing the saccade to face or non-face.

I greatly appreciate the authors' work for so many results. The data appear solid and the analyses make sense and are well explained in the main text and Methods. However, I note that most of the results are descriptive and on the brain-areas level, i.e. describe the whole-brain distribution of responses to various defined stimuli and events. Based on the current results, further statistical testing and story organization could improve the manuscript's novelty. Assuming these points can be addressed, I think this paper will make an important contribution to the field of visual exploration in humans and will be of interest to those who want to study cognitive neuroscience combined with advanced machine learning models from deep learning. Then I would recommend publication in NATURE COMMUNICATIONS.

Major comment

Most results are electrode ratios in brain regions (occipital, parietal, frontal, and temporal), and it is difficult to give precise localization of effects. I recommend two analytical methods to improve relevant results.

The ieeg database used in this manuscript has a considerable number of subjects and electrodes, so I recommend using an fMRI-like analysis method recently introduced in ieeg's whole-brain analysis, for example, Miller, et al, NATURE COMMUNICATIONS (2018). for your Figure 2b, Figure 3d, Figure 4d, Figure 5d, etc. At the same time, the results found by this new approach can refine the findings from the lobe level to the voxel level, making better connections with previous fMRI results, such as Baldassano, et al, neuron (2017), thereby improving the story organization of the manuscript.

The manuscript reported an anatomical gradient trend in some electrode distribution maps and a temporal gradient in TRF visually. However, it is a pity that no in-depth analysis is carried out. Reporting

the gradient characteristics of electrodes or brain regions will greatly enhance the contribution of this paper and build a link with many fields (See the review from Wang, Xiao-Jing, Nat Rev Neurosci (2020)).

I think of two candidate analysis ideas for the author's reference. One is recent work from Gao et al, eLife 2020, who invented a method and toolkit to estimate the gradient of the membrane time constant tau at electrodes/brain regions directly from the LFP data. Correlating the tau estimated from each electrode with the TRF, or comparing tau from the event electrode and others will more clearly reveal the relationship between the hierarchical gradient at the electrophysiological level and the semantic gradient at the cognitive level. Another is the recent work from Markello et al, Nature Methods 2022, who provided whole-brain gradient maps of various neural features, Figure 2 of their article, such as T1w/T2w, etc. Correlating these features with TRF can also reveal the relationship between hierarchical and semantic gradients.

Minor comments

Line 107: I suggest incorporating figure s1 into figure 1 to help readers quickly understand the definitions of three events of interest

Line 110-113 : These behavioral results, i.e. "increase" or "decrease" referred to in the text, lack a statistical test

Line133 : Figure 1d is missing the standard deviation over the group

Line140-143 : I appreciate the author's choice of frequency bands and the way of re-referencing

Line 179-181 : How this z-scoring been done? I did not see a description of this step in the methods section. If I understand correctly, you will get three TRFs on each electrode, corresponding to film cut, saccade, and motion if you merge these three and do a z-score. Then I think the positive and negative of TRF are only a relative relationship and cannot be interpreted as an increase or decrease in BHA. For example, the maximum amplitudes of the three TRFs are 1, 2, and 3. After the z-score, you interpret the first one as a BHA drop, which I don't think is reasonable.

Line 261 : Mixed linear models are used in several places in the manuscript. Looking at the data in Figure 3A, if a similar approach were used here, the parietal and frontal lobes might also be significant? other results include fig4a

Line402: Figure S20 first appeared in the Discussion section and seems like it should be mentioned first in the Results section

Line 556 : Better to draw electrode density maps and group density maps to visualize the electrode distribution in different brain regions, for example, Figure 2 from Miller, et al, NATURE COMMUNICATIONS (2018)

Line 592 : I think the time series of BHA needs to be baseline-corrected, such as doing a z-score with the mean and standard deviation of the whole data, to ensure that the magnitude of each subject and each electrode is similar.

Line 642 : How this parameter is determined. I found several smoothing parameters in the manuscript. I recommend that the authors give reasons for their choice, especially since the authors used BHA, which is a fast and transient signal.

Line 813 : typo. 1 should be 2

comments in Supplementary Materials

Line 83-88: I think Figure s5b is an interesting result. Please see the major comment

Line195 : " distance between two patches at random locations (green)", where is it ?

Reviewer #2 (Remarks to the Author):

In this study, the authors collected iEEG responses in 23 patients to more than 40 minutes of audiovisual movies. It takes a broad, exploratory approach to characterizing neural responses in terms of visual features of the movie (motion, film cuts, event transitions) and eye movements, and identifies a large number of electrodes exhibiting diverse responses to these features. Some of these features are straightforward to compute, while for others the authors introduce new machine-learning methods, such as for characterizing saccade novelty and face saccades.

This is a very impressive study, and is the first that has attempted to provide a comprehensive examination of intracranial responses to naturalistic movies. The analytic and statistical approaches all appear sound and appropriate, and the large size of the dataset (number of patients/channels and duration of recording) provide high power for characterizing temporal response functions throughout the brain. I can see these results serving as a starting point for many follow-up studies in the field, to examine the specific role of all of the regions identified in this work.

I reviewed a previous version of this paper at another journal, and was pleased to see that several of the issues I previously raised have been improved, including the visualizations of effects in Fig 2, interpretations of responses in auditory regions, and some clarifications of methodological details. These changes have further improved my initial positive assessment of the paper.

Minor comments:

1) The details of how event cuts and continuous cuts are defined should be clarified. The authors state that event cuts are "film cuts with the highest event salience above a change point" which was confusing to me - I believe this would be more simply stated as "film cuts with event salience above a threshold" followed by the description of how this threshold is defined using the change-point approach. Continuous cuts are then defined as low-salience cuts, but are also said to be matched to event cuts in terms of low-level features - how is this accomplished? Were these cuts naturally matched in terms of these features (i.e. the authors are reporting that there are no significant differences in these features) or is an explicit matching process used to ensure that event and continuous cuts do not differ on low-level properties?

2) I am still confused about the histograms in S14, even with the additional clarification in the caption. The leftmost non-zero bin appears to be very close to 0 (less than 20ms), and the caption says that the first saccade is removed - are there hundreds of cases in which there are multiple saccades in <20ms? Then, moving to the right, there are several bins exactly equal to zero, and then a bin with hundreds of saccades, followed by a couple more bins exactly equal to 0. Maybe this is somehow related to the log scaling - perhaps the binning was done on a linear scale and then stretched to a log scale, creating these gaps? Right around 140ms there is another strange gap, where a bin with 0 saccades is flanked by two bins each with >500 saccades. If this is just a data visualization issue then this is an easy fix, but if the saccades actually have this distribution (e.g. saccades were only observed at ~138ms and ~142ms but never 140ms) then this raises concerns about the details of how data was collected.

3) Unless I missed it somewhere in the supplementary tables, I didn't see the number of face vs non-face saccades reported. It would be useful to these to interpret the results - e.g. if non-face saccades are

relatively uncommon, that could explain why they are associated with large-magnitude signals (e.g. because only a highly-salient object can pull fixation away from faces).

Reviewer #3 (Remarks to the Author):

This study investigates neural responses to changes in visual input, specifically changes associated with 1) stimulus motion, 2) "film cuts" in videos, and 3) eye movements. Neural recordings from intracranial electrodes in humans were collected as participants passively watched videos. Responses related to stimulus motion were found primarily in low-level visual brain areas, while responses related to saccades and film cuts were observed more broadly across the brain. Responses in some locations in high-order brain areas were modulated by semantic novelty of the cut, and by novelty of the saccade target.

This is an impressive paper with a difficult-to-acquire dataset and extensive set of analyses. The question of how visual changes drive neural responses under realistic dynamic conditions is important; in simultaneously modeling visual changes due to stimulus motion, eye movements, and event-related cognitive transitions, the authors take on a challenge which is too often ignored in studies using movie stimuli. Confounds in the stimuli make the final results sometimes complicated or inconclusive, a well-known limitation of the naturalistic approach which trades off with the benefits of ecological validity. At times, the take-home messages of the paper come across as rather diffuse: for example, "semantic novelty is an important factor modulating neural responses to film", with effects often referred to as "widespread" across the brain. The paper would be improved by highlighting, and if possible strengthening, some of the specific concrete findings.

Major comments:

Throughout the paper there are references to various effects being especially prominent in "higher order" brain areas, but there is not much discussion of what is meant by "higher order". In some analyses, entire lobes are compared to each other; but it isn't established (as far as I know) that, for example, all of the temporal lobe is "higher" than all of the parietal lobe (this is implied in Figure 3, "Neural responses associated with film cuts are stronger, and in higher brain areas respond specifically to event cuts as opposed to continuous cuts"). Indeed, regions of the parietal lobe are in the default mode network and thus considered very "high" by some. The authors show results in smaller parcels (such as in Fig S12), but it isn't clear which parcels are considered "higher order". Quantification or further discussion of what is meant by "higher order" and which effects are observed in "higher order" brain areas is needed.

It is very interesting that different sets of channels were responsive to saccades with high and low semantic novelty (Fig 4); the authors' proposed explanation for this in the Discussion was that "multiple behaviors compete for eye movement control in natural environments". If an analysis could be performed to support this hypothesis, it would strengthen the paper.

Film cuts were categorized as either "event cuts" or "continuous cuts". "Event cuts" were film cuts which coincided with "narrative event boundaries" judged by humans, ie, "high salience" cuts. An equal number of "continuous cuts" were selected which had "low salience".

a. It seems that all film cuts which were not "high salience" were classified as "low salience", correct?

"Event cuts and continuous cuts were matched in changes of low-level visual features. These features are luminance, contrast, complexity, entropy, color, and features from layer fc7 of AlexNet".

b. What exactly is meant by "matched"? Presumably there is some limit to how well the cuts can be matched across categories, as the stimulus is not being modified. What was the procedure for selecting the continuous cuts? The number of event cuts and continuous cuts was matched for each movie; does this mean that some movies had better "match quality" than others? In general, readers should have enough information such that they could perform the same procedures on their own stimuli and dataset.

These points are important to clarify, given that the authors argued and demonstrated earlier in the paper that various kinds of visual changes drive widespread brain activity.

It is proposed that responses to motion in auditory cortex may have been due to an association between stimulus movement and sound. Can an analysis be performed to test this idea?

Minor comments:

There is a part C referenced in the caption of Figure S12, but it does not appear in the figure.

"There was little overlap in neural response associated with the main sources of visual changes analyzed here, except for film cuts and saccades, which may be due to unobserved common factors."

It is hard to understand this sentence.

We thank the reviewers for their substantial effort in providing extensive comments. We have addressed each one in detail below. The most significant change overall is that we have formalized the claim on “higher-order” brain areas. We followed the very useful examples the reviewers pointed us to in the literature with brain-wide analyses. The main finding is that the specificity increases along the cortical hierarchy from sensory to association areas defined by cortical myelination as measured by T1w/T2w ratio (Gao et al, 2020). Relatedly, we now also display many results on a standard atlas, rather than individual electrodes which were harder to visualize. Edits to the manuscript are indicated in blue, and all quoted here for convenience.

REVIEWER COMMENTS

Reviewer #1 (Remarks to the Author):

Summary

This study by Maximilian et al used intracranial EEG data in humans watching some movie clips and analyzed whole brain responses based on high gamma activity. 1. estimate temporal response function (TRF) of stimulus motion, film cut, and saccade and TRF's distribution across brain regions; 2. divide the film cut into event cut and continuous cut, and reveal this event boundary segmentation response in the whole brain; 3. Finally, the author focuses on the saccade in detail. On the one hand, by comparing the semantic distance of the pictures before and after the saccade to capture the electrode distribution of the semantic novelty. On the other hand, the distribution of social information novelty was analyzed by comparing the saccade to face or non-face.

I greatly appreciate the authors' work for so many results. The data appear solid and the analyses make sense and are well explained in the main text and Methods. However, I note that most of the results are descriptive and on the brain-areas level, i.e. describe the whole-brain distribution of responses to various defined stimuli and events. Based on the current results, further statistical testing and story organization could improve the manuscript's novelty. Assuming these points can be addressed, I think this paper will make an important contribution to the field of visual exploration in humans and will be of interest to those who want to study cognitive neuroscience combined with advanced machine learning models from deep learning. Then I would recommend publication in NATURE COMMUNICATIONS.

Major comment

Most results are electrode ratios in brain regions (occipital, parietal, frontal, and temporal), and it is difficult to give precise localization of effects. I recommend two analytical methods to improve relevant results.

The i EEG database used in this manuscript has a considerable number of subjects and electrodes, so I recommend using an fMRI-like analysis method recently introduced in i EEG's whole-brain analysis, for example, Miller, et al, NATURE COMMUNICATIONS (2018). for your Figure 2b, Figure 3d, Figure 4d, Figure 5d, etc. At the same time, the results found by this new approach can refine the findings from the lobe level to the voxel level, making better connections with

previous fMRI results, such as Baldassano, et al, neuron (2017), thereby improving the story organization of the manuscript.

This was great advice. Thank you. Instead of showing results on individual electrodes (that are hard to localize visually) we now map electrodes to the cortical surface and use the Glasser and Desikan-Killiany parcelations of brain areas (Glasser et al. 2016; Desikan et al. 2006). Not only did this improve visualization, but it also refined our analysis of spatial effects to obtain more fine grained results. These updates are reflected in the current Figures 1A, 2B, 3B&E, 4B&E, 5B&E, S4, S5, S13, S18 and S21. We followed the approach by Gao et al., 2020 and grouped responses into finer grained parcels of functionally or anatomically defined cortical atlases. This allows for the comparison of our results with previous studies, as well as correlations with anatomical and functional gradients. The comparison to previous results is particularly fruitful in the example of response to different types of film cuts, as described in the discussion: “Our results show that event cuts elicit stronger responses (than continuous cuts) in association areas like the angular gyrus and posterior medial cortex (Figure 3B), consistent with previous fMRI results (Baldassano et al. 2017). We find the opposite effect in the superior parietal and premotor cortex (Figure 3B), similar to a previous fMRI study (Magliano and Zacks 2011). This is consistent with the suggestion that the superior parietal cortex maintains internal sensory and motor representations (Wolpert, Goodbody, and Husain 1998).”

An introduction to the new approach is found in the section “Neural responses associated with stimulus motion, saccades and film cuts”: “To map responsive channels to the cortical surface we compute a weighted average of the response amplitude in each parcel of the Glasser atlas (Glasser et al. 2016) following a method described by Gao et al. (Gao et al. 2020) (Figure 2B).”

Further details are described in “Cuts at event boundaries are associated with distinct neural responses across the brain”: “We compute the weighted average of the amplitude difference of event versus continuous cuts across all channels in each parcel of the Glasser atlas to visualize this effect on the cortical surface (Figure 3B) (Glasser et al. 2016; Gao et al. 2020). This analysis reveals a complex pattern of brain areas responding preferentially either to event or continuous cuts.

Section “Novelty across saccades”: “An analysis of the amplitude difference in smaller parcels shows that most of the cortex responds stronger to high-novelty saccades. However, there are some exceptions, particularly in the parietal and temporal lobes, responding stronger to low-novelty saccades (Figure 4B).”

Section “Saccades to faces”: “Consistent with this, computing the weighted average of the amplitude difference of responses confirms stronger responses to non-face saccades in most brain areas (Figure 5B). However, this fine grained analysis also shows some areas in the parietal and temporal lobes with larger responses to face saccades (Figure 5B)”

The manuscript reported an anatomical gradient trend in some electrode distribution maps and a temporal gradient in TRF visually. However, it is a pity that no in-depth analysis is carried out. Reporting the gradient characteristics of electrodes or brain regions will greatly enhance the contribution of this paper and build a link with many fields (See the review from Wang, Xiao-Jing, Nat Rev Neurosci (2020)).

I think of two candidate analysis ideas for the author's reference. One is recent work from Gao et al, eLife 2020, who invented a method and toolkit to estimate the gradient of the membrane time constant tau at electrodes/brain regions directly from the LFP data. Correlating the tau estimated from each electrode with the TRF, or comparing tau from the event electrode and others will more clearly reveal the relationship between the hierarchical gradient at the electrophysiological level and the semantic gradient at the cognitive level. Another is the recent work from Markello et al, Nature Methods 2022, who provided whole-brain gradient maps of various neural features, Figure 2 of their article, such as T1w/T2w, etc. Correlating these features with TRF can also reveal the relationship between hierarchical and semantic gradients.

Thank you for this suggestion. Gao et al, 2020 and Markello et al 2022 provide excellent resources that allow us to link our results to anatomical and functional gradients. New results are reported in Figures 3E, 4E, 5E, S13, S18 and S21. In particular, we were able to show that the specificity of responses to event versus continuous cuts increase along the cortical hierarchy defined by cortical myelination measured by the T1w/T2w ratio (Wang, 2020). This hierarchy leads from sensory to association areas (Gao et al, 2020). The specificity of responses to high or low novelty saccades also follows this hierarchy. This provides statistical evidence and conceptual precision to the qualitative observation we had made that specificity increases from “lower level” to “higher level” brain areas. In addition we tested the association of our results with the functional gradient defined in Margulies et al 2016. We found that specificity is associated with gradient 2, which has been reported to separate somatomotor and auditory cortex, while they are unrelated to gradient 1 spanning from primary sensorimotor to transmodal regions (Figures S13, S18 and S21).

We report associations of specificity with anatomical and functional gradients in parcels of the coarser Desikan-Killiany atlas. We needed this coarser atlas to have a sufficient number of responsive channels to compute “specificity”. However, the same analysis in smaller parcels of the Glasser atlas confirms our results (Figures S13, S18 and S21) albeit with many of the smaller parcels providing no data on specificity.

The results in Figure 2A we provide an example for the variety of time courses of BHA responses. We now summarize these results across the brain in Figure 2B using the Glasser atlas for visualization purposes only, and leave more detailed analysis for subsequent figures.

The new methods are described in the methods section as follows:

“Registration of responses to cortical atlases

In order to localize effects to specific anatomically or functionally defined brain areas we adapt the pipeline developed by Gao et al. (Gao et al. 2020) (<https://github.com/rdgao/field-echos>). Electrode coordinates from the iElvis pipeline described above are transformed from MNI305 to MNI152 space with an affine transformation (<https://surfer.nmr.mgh.harvard.edu/fswiki/CoordinateSystems>). In Figure 1A electrodes within 4 mm distance to a parcel of the Glasser atlas are counted. For maps of weighted average responses in Figures 2B, 3B, 4B, 5B and S4A we follow the approach described in Gao et al. (Gao et al. 2020). Briefly, responses of all electrodes close to a parcel are weighted by distance following a Gaussian with full-width-half-max of 4 mm. Parcels where the highest weighted count is below 0.5 are excluded from analysis (gray parcels in plots). The pipeline in Gao et al has been implemented with the Glasser atlas (Glasser et al. 2016). When analyzing specificity, responsive electrodes are sparse. To address this we utilize the coarser Desikan-Killiany atlas (Figures 3E, 4E, 5E, S13, S18 and S21). Voxel resolution T1w/T2w ratio data is available through the neuromaps toolbox (Markello et al. 2022). We transform this data onto the freesurfer surface using the neuromaps toolbox. This enables us to average T1w/T2w data within parcels of the Desikan-Killiany atlas available through freesurfer (Desikan et al. 2006). We follow the same approach to obtain functional gradients in the Glasser and Desikan-Killiany parcellation (Figures S13, S18, S21).

Correlation to gradients

Registering results to standardized brain maps allows correlation of our results to anatomical and functional gradients. For brain maps in the coarser Desikan-Killiany atlas we compute the Spearman correlation coefficient and the associated p-value assuming that these larger parcelations provide independent measures. For brain maps in the finer Glasser atlas we follow Gao et al. and compute significance using permutation statistics taking into consideration spatial autocorrelation (Gao et al. 2020)."

We describe our new results the following sections

- 1) "Cuts at event boundaries are associated with distinct neural responses across the brain": "To directly test whether specificity increases in higher-order brain areas, we compute the spatial correlation of specificity with maps of cortical hierarchy (Figure 3E). In particular, gray matter myelination, measured by the T1w/T2w ratio, has been suggested as an indirect measure of hierarchy from sensory to association processing (Gao et al. 2020; Wang 2020). We find a significant negative correlation of specificity with the T1w/T2w ratio (Figure 3E, right) indicative of increased specificity along the sensory to association hierarchy (Gao et al. 2020; Wang 2020). This correlation holds when the analysis is conducted on a more fine-grained atlas and when testing significance with permutation statistics accounting for the spatial autocorrelation of cortical myelination across adjacent brain areas (Figure S13). In addition, specificity is negatively correlated with the second principal gradient of functional connectivity, which separates somatomotor

and auditory cortices (Figure S13) (Margulies et al. 2016). There is no consistent association of specificity to the first principal gradient of functional connectivity capturing sensorimotor-to-transmodal progression (Figure S13) (Margulies et al. 2016)."

- 2) "Novelty across saccades": "The comparison of specificity with T1w/T2w ratio leads to strong negative correlations indicating that specificity increased along the sensory to association hierarchy (Figure 4E & S18A). In addition, specificity of responses to high- and low-novelty saccades is related to the second gradient of functional connectivity, separating somatomotor and auditory cortices (Figure S18C). There was no consistent association with the first gradient of functional connectivity (Figure S18B). As such, these results parallel those for specificity to type of cuts."
- 3) "Saccades to faces": "The spatial correlation of specificity with the T1w/T2w map is not significant for larger parcels in the Desikan-Killiany atlas (Figure 5E). This result is unsurprising given that responses across the brain are specific to non-face saccades starting from sensory areas (Figure 5B&D). There is a significant correlation, however, of the specificity to face and non-face saccades and the second functional gradient, which separates somatomotor and auditory cortices (Figure S21)."

The implications of our results are summarized in the discussion section: "We show that event boundaries are processed in a widespread ensemble with increased specificity in higher-order association areas (Figure 3E)."

Minor comments

Line 107: I suggest incorporating figure s1 into figure 1 to help readers quickly understand the definitions of three events of interest

This figure is now incorporated into panel C of Figure 1.

Line 110-113 : These behavioral results, i.e. "increase" or "decrease" referred to in the text, lack a statistical test

We now establish the significance of saccade probability in time bins relative to film cuts against a surrogate distribution of saccade probability. This surrogate distribution is created by random circular shuffles of the time of film cuts. Time bins with significant saccade probability compared to the mean are now marked with stars in Figure 1D.

Line133 : Figure 1d is missing the standard deviation over the group

In contrast to panels E and F, panel D is based on counts of saccades in time bins around cuts. The standard error of the mean here is calculated based on a poisson distribution as $\sqrt{\lambda}/n$, where lambda is the count of saccades in a specific bin and n the total number of saccades. In this case, with 32,942 saccades the error is about 5 orders of magnitude

smaller than the count of saccades in each bin. We think that adding these error bars would not add much to the figure, but hope that the permutation statistics as described above will add confidence to the results.

Line140-143 : I appreciate the author's choice of frequency bands and the way of re-referencing

Thank you!

Line 179-181 : How has this z-scoring been done? I did not see a description of this step in the methods section. If I understand correctly, you will get three TRFs on each electrode, corresponding to film cut, saccade, and motion if you merge these three and do a z-score. Then I think the positive and negative of TRF are only a relative relationship and cannot be interpreted as an increase or decrease in BHA. For example, the maximum amplitudes of the three TRFs are 1, 2, and 3. After the z-score, you interpret the first one as a BHA drop, which I don't think is reasonable.

Thank you for pointing out the lack of clarity. Z-scoring is used only to better visualize the shape of the different waveforms in Figure 2A-top. It is not used anywhere else in the paper. For that figure only, the z-score is computed individually for each TRF separately. All other analyses of amplitude and significance operate on the un-normalized TRFs: “For visualization of the time-course in Fig. 2A-top, each TRF is z-scored and baseline corrected individually for each channel and each regressor (also corresponding panel in S3, S11, S16). The baseline is computed between 2.1s and 3s after stimulus onset (film cuts, saccades or motion). Electrodes are grouped in clusters of similar waveforms (Yeo et al. 2011; Lashkari et al. 2010). For Figure 2A-bottom we average similar TRF waveforms in the same cluster. Here, the scale is the same across all regressors for better comparison. For Figure 2B we compute the maximum amplitude across time (preserving the sign) and average over all channels within a parcel of the Glasser atlas (Glasser et al. 2016). Positive values (red) indicate a dominant enhancement of TRF in that area. Negative values (blue) indicate a dominant suppression. Zero values (white) indicate that enhancement and suppression are similarly strong.”

Line 261 : Mixed linear models are used in several places in the manuscript. Looking at the data in Figure 3A, if a similar approach were used here, the parietal and frontal lobes might also be significant? other results include fig4a

Thank you for pointing this out. We used a mixed-design ANOVA in most cases. We have updated the manuscript to clarify this:

Results section “Cuts at event boundaries are associated with distinct neural responses across the brain”: “A follow-up mixed-design ANOVA in each brain region with patient as random factor shows that amplitudes are significantly larger for event cuts in the temporal lobe and MTL (Figure 3A, temporal: $F(1,803) = 48.5$, $p = 7 \times 10^{-12}$; MTL: $F(1,95) = 23.2$, $p = 5.4 \times 10^{-6}$; FDR correction at $q = 0.05$, full results in Table S4).”

Caption of Figure 3: “Stars indicate significant main effects of the type of film cut within regions with patients as a random factor in a mixed-design ANOVA (FDR corrected at $q = 0.05$, Table S4).”

Results section: “Novelty across saccades”: “A follow-up mixed-design ANOVA for each brain region show that novelty increases response amplitude in occipital and temporal lobes (Figure 4A, occipital: $F(1,264) = 9.06$, $p = 0.0043$, temporal $F(1,513) = 11.83$, $p = 0.0011$, FDR corrected at $q = 0.05$, Table S7).”

Caption of Figure 4: “Stars indicate significant main effects of the type of film cut within regions with patients as a random factor in a mixed-design ANOVA (FDR corrected at $q = 0.05$, Table S7).”

Results section “Saccades to faces”: “[...] (follow-up mixed-design ANOVA for each brain region with patient as random effect, Figure 5A: [...])”

Caption of Figure 5: Stars indicate significant main effects of the type of film cut within regions with patients as a random factor in a mixed-design ANOVA (FDR corrected at $q = 0.05$, Table S10).

Line402: Figure S20 first appeared in the Discussion section and seems like it should be mentioned first in the Results section

To improve organization of the manuscript we have moved the description of face motion from the Discussion to the Results section (“Face Motion”). These results are confirmatory of previous literature.

Line 556 : Better to draw electrode density maps and group density maps to visualize the electrode distribution in different brain regions, for example, Figure 2 from Miller, et al, NATURE COMMUNICATIONS (2018)

Figure 1A has been updated and now shows the number of electrodes in different brain regions as defined by the glasser atlas (Glasser et al. 2016). We adapted code from Gao et al. 2020 (Gao et al. 2020) to localize and count electrodes within 4 mm of each parcel. This is also described in the new methods section “Registration of responses to cortical atlases”: “In Figure 1A electrodes within 4 mm distance to a parcel of the Glasser atlas are counted.”

Line 592 : I think the time series of BHA needs to be baseline-corrected, such as doing a z-score with the mean and standard deviation of the whole data, to ensure that the magnitude of each subject and each electrode is similar.

We do z-score as described above, mostly for visualization across electrodes and patients (Figure 2A). Note that significance of response is established within each channel against surrogate distributions of TRFs. As such, significance is insensitive to z-scoring. The estimation of amplitude difference in Figures 3A, 4A and 5A are based on fits of individual events to TRFs within each channel. As we evaluate amplitude ratios (log-difference) this analysis is also independent of overall scale.

Line 642 : How this parameter is determined. I found several smoothing parameters in the manuscript. I recommend that the authors give reasons for their choice, especially since the authors used BHA, which is a fast and transient signal.

While BHA is a fast signal, event boundaries are annotated manually with less accurate timing. Previous studies have aggregated event boundaries in 1.5-2s windows (Jafarpour et al. 2019; Ben-Yakov and Henson 2018), roughly in the range of our smoothing parameter for these behavioral time markers (0.5s standard deviation of a Gaussian). We use event saliency obtained with this smoothing to classify film cuts as event or continuous cuts, not directly to analyze neural responses. This approach is visualized in Figure S7A. Part of the motivation to use cuts in our analysis and not annotations of event boundaries or saliency is to make use of their higher temporal precision. These work for fMRI analysis with lower temporal resolution, but not to analyze fast signals as the BHA.

To clarify this issue, we refer to event saliency as vector not regressor, because it is not used directly in the analysis. Further we add a reason for the choice of the smoothing parameter: “This vector was then smoothed with a Gaussian of 0.5s, to account for variance in the manual annotation of event boundaries (Figure S7A) (Ben-Yakov and Henson 2018; Jafarpour et al. 2019).”

Another smoothing parameter is applied on the TRFs before cluster correction. We added our motivation for this choice in the section “System identification approach to establish temporal response functions.”: “After estimation, TRFs are smoothed with a Gaussian window with a standard deviation of 53ms to filter higher-frequency noise. Similar smoothing is common in cluster-based statistical analysis often performed in fMRI research and increases sensitivity of the analysis (Friston et al. 1996; Chumbley and Friston 2009).”

Line 813 : typo. 1 should be 2

This was corrected: “Assume N_1 channels respond to condition 1, N_2 channels respond to condition 2, [...]”

comments in Supplementary Materials

Line 83-88: I think Figure s5b is an interesting result. Please see the major comment

This is indeed an interesting result. We now highlight the main observation on timing in the results section “Neural responses associated with stimulus motion, saccades and film cuts”: “TRFs vary widely in terms of amplitude, duration and onset across the brain (Figure 2, S3, S4 & S5) [...]. The onset of responses associated with film cuts predominantly starts after the cuts, while in the case of motion the onset occurs before the stimulus (Figure

S5). The onset of responses associated with saccades varies widely across the brain with onsets before saccades in some, and after in other areas (Figure S5).”

However, we decided to focus on the location and specificity of responses, and leave analysis of the timing of responses for future studies. As it is, the paper is already quite extensive.

Line195 : ” distance between two patches at random locations (green)”, where is it ?

Thanks for pointing this out, it referred to an earlier version of the figure and has been removed.

Reviewer #2 (Remarks to the Author):

In this study, the authors collected iEEG responses in 23 patients to more than 40 minutes of audiovisual movies. It takes a broad, exploratory approach to characterizing neural responses in terms of visual features of the movie (motion, film cuts, event transitions) and eye movements, and identifies a large number of electrodes exhibiting diverse responses to these features. Some of these features are straightforward to compute, while for others the authors introduce new machine-learning methods, such as for characterizing saccade novelty and face saccades.

This is a very impressive study, and is the first that has attempted to provide a comprehensive examination of intracranial responses to naturalistic movies. The analytic and statistical approaches all appear sound and appropriate, and the large size of the dataset (number of patients/channels and duration of recording) provide high power for characterizing temporal response functions throughout the brain. I can see these results serving as a starting point for many follow-up studies in the field, to examine the specific role of all of the regions identified in this work.

I reviewed a previous version of this paper at another journal, and was pleased to see that several of the issues I previously raised have been improved, including the visualizations of effects in Fig 2, interpretations of responses in auditory regions, and some clarifications of methodological details. These changes have further improved my initial positive assessment of the paper.

Minor comments:

1) The details of how event cuts and continuous cuts are defined should be clarified. The authors state that event cuts are "film cuts with the highest event salience above a change point" which was confusing to me - I believe this would be more simply stated as "film cuts with event salience above a threshold" followed by the description of how this threshold is defined using the change-point approach. Continuous cuts are then defined as low-salience cuts, but are also said to be matched to event cuts in terms of low-level features - how is this accomplished? Were these cuts naturally matched in terms of these features (i.e. the authors are reporting that there are no

significant differences in these features) or is an explicit matching process used to ensure that event and continuous cuts do not differ on low-level properties?

We have clarified our definition of event and continuous cuts as well as the matching procedure in the methods section:

“Film cuts were sorted by event salience. ‘Event cuts’ are the film cuts with event salience above a threshold. For each movie this threshold is defined as the change point detected with the `findchangepts()` function in MATLAB. This method [...]. We select 57 event cuts out of a total of 561 cuts (Table S2). An equal number of ‘Continuous cuts’ is selected from cuts with the lowest event salience. We test if event cuts and continuous cuts differ in changes of low-level visual features (Zheng et al. 2021). These features are [...]. We compute a p-value for each feature across cuts with a Wilcoxon signed rank test and correct for multiple comparisons with the Benjamini-Hochberg procedure. If event and continuous cuts differ in low-level features we iteratively select a random set of continuous cuts from the film cuts in the lower half of event salience until there is no difference in low-level features.”

2) I am still confused about the histograms in S14, even with the additional clarification in the caption. The leftmost non-zero bin appears to be very close to 0 (less than 20ms), and the caption says that the first saccade is removed - are there hundreds of cases in which there are multiple saccades in <20ms? Then, moving to the right, there are several bins exactly equal to zero, and then a bin with hundreds of saccades, followed by a couple more bins exactly equal to 0. Maybe this is somehow related to the log scaling - perhaps the binning was done on a linear scale and then stretched to a log scale, creating these gaps? Right around 140 ms there is another strange gap, where a bin with 0 saccades is flanked by two bins each with >500 saccades. If this is just a data visualization issue then this is an easy fix, but if the saccades actually have this distribution (e.g. saccades were only observed at ~138ms and ~142ms but never 140ms) then this raises concerns about the details of how data was collected.

Thanks for pointing this out. It is in fact related to the binning. The data was binned on a log scale, but given the 60Hz sampling of the underlying data there are gaps at short time scales. The histograms are now shown as curves effectively omitting these artifactual gaps (Figure S15).

3) Unless I missed it somewhere in the supplementary tables, I didn't see the number of face vs non-face saccades reported. It would be useful to these to interpret the results - e.g. if non-face saccades are relatively uncommon, that could explain why they are associated with large-magnitude signals (e.g. because only a highly-salient object can pull fixation away from faces).

For each recording we select an equal number of face and non-face saccades. This is to make sure the TRFs are not biased by the number of events. Across all recordings and patients we analyzed 10,225 face and non-face saccades each. This number is now mentioned in the results section “Saccades to faces”: “This network was then used to

detect faces in the unlabeled frames. We then divide saccades to faces and saccades to other objects (non-face saccades, 10,225 each) [...]"

Reviewer #3 (Remarks to the Author):

This study investigates neural responses to changes in visual input, specifically changes associated with 1) stimulus motion, 2) "film cuts" in videos, and 3) eye movements. Neural recordings from intracranial electrodes in humans were collected as participants passively watched videos. Responses related to stimulus motion were found primarily in low-level visual brain areas, while responses related to saccades and film cuts were observed more broadly across the brain. Responses in some locations in high-order brain areas were modulated by semantic novelty of the cut, and by novelty of the saccade target.

This is an impressive paper with a difficult-to-acquire dataset and extensive set of analyses. The question of how visual changes drive neural responses under realistic dynamic conditions is important; in simultaneously modeling visual changes due to stimulus motion, eye movements, and event-related cognitive transitions, the authors take on a challenge which is too often ignored in studies using movie stimuli. Confounds in the stimuli make the final results sometimes complicated or inconclusive, a well-known limitation of the naturalistic approach which trades off with the benefits of ecological validity. At times, the take-home messages of the paper come across as rather diffuse: for example, "semantic novelty is an important factor modulating neural responses to film", with effects often referred to as "widespread" across the brain. The paper would be improved by highlighting, and if possible strengthening, some of the specific concrete findings.

Major comments:

Throughout the paper there are references to various effects being especially prominent in "higher order" brain areas, but there is not much discussion of what is meant by "higher order". In some analyses, entire lobes are compared to each other; but it isn't established (as far as I know) that, for example, all of the temporal lobe is "higher" than all of the parietal lobe (this is implied in Figure 3, "Neural responses associated with film cuts are stronger, and in higher brain areas respond specifically to event cuts as opposed to continuous cuts"). Indeed, regions of the parietal lobe are in the default mode network and thus considered very "high" by some. The authors show results in smaller parcels (such as in Fig S12), but it isn't clear which parcels are considered "higher order". Quantification or further discussion of what is meant by "higher order" and which effects are observed in "higher order" brain areas is needed.

Thank you for pointing out this issue. Following similar suggestions of reviewer #1 we have correlated the effects reported in our paper with anatomical and functional gradients. In particular the specificity of responses to different types of film cuts and saccades correlated with the T1w/T2w ratio. The T1w/T2w ratio is a proxy of gray matter myelination and has been suggested to reflect a cortical hierarchy from sensory to association regions

(Wang et al. 2020, Gao et al. 2020). We now define order along this hierarchy, 'higher-order' brain areas thus are association areas with a lower T1wT2w ratio.

We introduce this concept in the introduction: "Higher-order association areas are defined as those that lie higher on the sensory-association hierarchy, as defined by cortical microstructure (Gao et al. 2020)."

In the rest of the manuscript we refer to "higher-order association areas" to describe brain areas higher on the sensory-to-association hierarchy. We have used this to establish the relationship quantitatively in Figure 3E, 4E and 5E.

It is very interesting that different sets of channels were responsive to saccades with high and low semantic novelty (Fig 4); the authors' proposed explanation for this in the Discussion was that "multiple behaviors compete for eye movement control in natural environments". If an analysis could be performed to support this hypothesis, it would strengthen the paper.

Thank you for this comment. To investigate this hypothesis we now tested changes in saccade novelty and amplitude in a repeated presentation that was available for one of the videos ('The Present'). Repeated presentation of visual stimuli changes exploration behavior. In images this is reflected by a decrease in saccade amplitude (Lancry-Dayan, Kupershmidt, and Pertzov 2019). In our data saccade amplitude increases during the second repeat of 'The Present'. In contrast to previous findings in images this would indicate an increase in exploratory behavior during repeated presentation of a stimulus. In our interpretation, this is consistent with the finding that In dynamic videos, eye movements are synchronized across subjects (Dorr et al. 2010; Madsen and Parra 2022). In the second presentation we typically observe a drop in ISC, suggesting a less stereotypical and more idiosyncratic exploratory behavior. During the first presentation of a video eye gaze is guided by common features that attract attention (Madsen and Parra 2022). After gaining familiarity with the narrative, a second presentation of a video allows for idiosyncratic visual exploration. While exploratory behavior for repeated video changed differently than for repeated static images, it is clear that the task changes the balance between stereotypical versus exploratory saccade behavior.

These new results are described in the results section "Novelty across saccades": "Behavioral data shows that saccade novelty does not significantly differ between repeats of "The Present" when controlling for saccade amplitude (difference of median novelty: $-7 \cdot 10^{-4}$, $p=0.46$, $N_{\text{repeat}_1}=6,843$, $N_{\text{repeat}_2}=5,121$, Wilcoxon rank sum test). However, saccade amplitude increases significantly in the second repeat of "The Present" (difference of median amplitude 0.55 DVA, $p=1.5 \cdot 10^{-31}$, $N_{\text{repeat}_1}=6,843$, $N_{\text{repeat}_2}=5,121$, Wilcoxon rank sum test)."

The implications are described in the discussion: "In static natural scenes, eye movements target objects with similar semantics (Wu, Wang, and Pomplun 2014). We propose that this type of saccades corresponds to saccades with low semantic novelty in our video

data. Saccades are also attracted by visual features that are novel in the context of static scenes (Ernst, Becker, and Horstmann 2020). We propose that this corresponds to saccades with high semantic novelty in our data. Exploitation of a visual scene by sampling similar objects competes with the necessity to explore novel objects in dynamic environments. We suggest that these objectives are concurrent and the balance may vary with task demands. For instance, we observed that saccade amplitude, a measure of exploratory behavior, increases during the second repeat of the video "The Present". In movies, eye movements are synchronized between subjects, suggesting a stereotypical behavior (Dorr et al. 2010; Madsen and Parra 2022). Increased saccade amplitude upon repeat could therefore indicate idiosyncratic exploration (Madsen and Parra 2022). Interestingly the opposite behavior is found for static images which are less synchronized across subjects (Dorr et al. 2010) and results in shorter saccades upon repetition (Lancry-Dayana, Kupershmidt, and Pertzov 2019). Together this suggests that the balance between exploitation and exploration depend on the task. Indeed, we find that after film cuts, saccades are directed primarily to novel targets (Figure S15). We suggest that neural responses to high-semantic novelty saccades support exploration, and responses to low-semantic novelty saccades support exploitation. The concurrent demands are reflected in different sets of channels that are responsive to saccades with high and low semantic novelty (Figure 4). The suppression of neural processing in the parietal and frontal lobes that we find in particular for low-semantic novelty saccades (Figure S16), may facilitate perceptual stability across eye movements (Ibbotson and Krekelberg 2011)."

Film cuts were categorized as either "event cuts" or "continuous cuts". "Event cuts" were film cuts which coincided with "narrative event boundaries" judged by humans, ie, "high salience" cuts. An equal number of "continuous cuts" were selected which had "low salience".

a. It seems that all film cuts which were not "high salience" were classified as "low salience", correct?

Sorry, this was not clearly worded. Hopefully this is clearer now: An equal number of cuts with lowest salience were selected, which we refer to as 'Continuous cuts'

"Event cuts and continuous cuts were matched in changes of low-level visual features. These features are luminance, contrast, complexity, entropy, color, and features from layer fc7 of AlexNet".

b. What exactly is meant by "matched"? Presumably there is some limit to how well the cuts can be matched across categories, as the stimulus is not being modified. What was the procedure for selecting the continuous cuts? The number of event cuts and continuous cuts was matched for each movie; does this mean that some movies had better "match quality" than others? In general, readers should have enough information such that they could perform the same procedures on their own stimuli and dataset.

These points are important to clarify, given that the authors argued and demonstrated earlier in the paper that various kinds of visual changes drive widespread brain activity.

Similar concerns have been expressed by reviewer #2. The methods section has been updated to clarify the selection of matching low salience continuous cuts:

“Film cuts were sorted by event salience. ‘Event cuts’ are the film cuts with event salience above a threshold. For each movie this threshold is defined as the change point detected with the findchangepts() function in MATLAB. This method [...]. We select 57 event cuts out of a total of 561 cuts (Table S2). An equal number of cuts with lowest salience were selected, which we refer to as ‘Continuous cuts’. We test if event cuts and continuous cuts differ in changes of low-level visual features (Zheng et al. 2021). These features are [...]. We compute a p-value for each feature across cuts with a Wilcoxon signed rank test and correct for multiple comparisons with the Benjamini-Hochberg procedure. If event and continuous cuts differ in low-level features we iteratively select a random set of continuous cuts from the film cuts in the lower half of event salience until there is no difference in low-level features.”

It is proposed that responses to motion in the auditory cortex may have been due to an association between stimulus movement and sound. Can an analysis be performed to test this idea?

Yes. To support this statement we have correlated the sound envelope from the movie stimuli with the optical flow signal used in our analysis. We find a significant correlation ($r = 0.24$, $p < 10^{-4}$, permutation statistics). Since the TRF analysis measures linear associations between optical flow and the BHA, this suggests that part of the responses to motion can be explained by the sound envelope.

We added results of this correlation to the section “Neural responses associated with stimulus motion, saccades and film cuts”: “For instance, sound is associated with motion (\$r=0.24\$, \$p < 10^{-4}\$, permutation statistics). This partly explains the responses associated with motion in Heschl’s gyrus, which is considered an auditory processing area.”

And in the “Limitations” section: “For instance, stimulus movement is associated with sound, such that responses to movement *likely* drive auditory areas.”

The approach is described in the methods section “Stimulus motion”: “For movies with sound (“Despicable Me English”, “Despicable Me Hungarian”, “The Present”) we correlate the motion regressor with the sound envelope. The sound envelope is calculated as the absolute value of the Hilbert transform of the sound extracted from the movie files. The correlation is computed across the concatenated signal for all three movies. To test the significance of this correlation we compute a surrogate distribution of the motion regressor by 10,000 circular shuffles.”

Minor comments:

There is a part C referenced in the caption of Figure S12, but it does not appear in the figure.

The caption has been removed.

"There was little overlap in neural response associated with the main sources of visual changes analyzed here, except for film cuts and saccades, which may be due to unobserved common factors."

It is hard to understand this sentence.

This sentence has been clarified as: "In most recording locations we found responses to only one of the main sources of visual change (Figure 2C). However, some locations respond to both film cuts and saccades, which may be due to unobserved common factors."

References

- Baldassano, Christopher, Janice Chen, Asieh Zadbood, Jonathan W. Pillow, Uri Hasson, and Kenneth A. Norman. 2017. "Discovering Event Structure in Continuous Narrative Perception and Memory." *Neuron* 95 (3): 709-721.e5. <https://doi.org/10.1016/j.neuron.2017.06.041>.
- Ben-Yakov, Aya, and Richard N. Henson, eds. 2018. "The Hippocampal Film Editor: Sensitivity and Specificity to Event Boundaries in Continuous Experience." *The Journal of Neuroscience* 38 (47): 10057–68. <https://doi.org/10.1523/JNEUROSCI.0524-18.2018>.
- Chumbley, J, and K Friston. 2009. "False Discovery Rate Revisited: FDR and Topological Inference Using Gaussian Random Fields." *NeuroImage* 44 (1): 62–70. <https://doi.org/10.1016/j.neuroimage.2008.05.021>.
- Desikan, Rahul S., Florent Ségonne, Bruce Fischl, Brian T. Quinn, Bradford C. Dickerson, Deborah Blacker, Randy L. Buckner, et al. 2006. "An Automated Labeling System for Subdividing the Human Cerebral Cortex on MRI Scans into Gyral Based Regions of Interest." *NeuroImage* 31 (3): 968–80. <https://doi.org/10.1016/j.neuroimage.2006.01.021>.
- Dorr, Michael, Thomas Martinetz, Karl R. Gegenfurtner, and Erhardt Barth. 2010. "Variability of Eye Movements When Viewing Dynamic Natural Scenes." *Journal of Vision* 10 (10): 28–28. <https://doi.org/10.1167/10.10.28>.
- Ernst, Daniel, Stefanie Becker, and Gernot Horstmann. 2020. "Novelty Competes with Saliency for Attention." *Vision Research* 168 (March): 42–52. <https://doi.org/10.1016/j.visres.2020.01.004>.
- Friston, K. J., A. Holmes, J-B. Poline, C. J. Price, and C. D. Frith. 1996. "Detecting Activations in PET and FMRI: Levels of Inference and Power." *NeuroImage* 4 (3): 223–35. <https://doi.org/10.1006/nimg.1996.0074>.
- Gao, Richard, Ruud L van den Brink, Thomas Pfeffer, and Bradley Voytek. 2020. "Neuronal Timescales Are Functionally Dynamic and Shaped by Cortical Microarchitecture." Edited by Martin Vinck, Laura L Colgin, and Thilo Womelsdorf. *ELife* 9 (November): e61277. <https://doi.org/10.7554/eLife.61277>.
- Glasser, Matthew F., Timothy S. Coalson, Emma C. Robinson, Carl D. Hacker, John Harwell, Essa Yacoub, Kamil Ugurbil, et al. 2016. "A Multi-Modal Parcellation of Human Cerebral Cortex." *Nature* 536 (7615): 171–78. <https://doi.org/10.1038/nature18933>.
- Ibbotson, Michael, and Bart Krekelberg. 2011. "Visual Perception and Saccadic Eye Movements." *Current Opinion in Neurobiology*, Sensory and motor systems, 21 (4): 553–58. <https://doi.org/10.1016/j.conb.2011.05.012>.
- Jafarpour, Anna, Sandon Griffin, Jack J. Lin, and Robert T. Knight. 2019. "Medial Orbitofrontal Cortex, Dorsolateral Prefrontal Cortex, and Hippocampus Differentially Represent the

- Event Saliency." *Journal of Cognitive Neuroscience* 31 (6): 874–84.
https://doi.org/10.1162/jocn_a_01392.
- Lancry-Dayana, Oryah C., Ganit Kupershmidt, and Yoni Pertzov. 2019. "Been There, Seen That, Done That: Modification of Visual Exploration across Repeated Exposures." *Journal of Vision* 19 (12): 2. <https://doi.org/10.1167/19.12.2>.
- Lashkari, Danial, Ed Vul, Nancy Kanwisher, and Polina Golland. 2010. "Discovering Structure in the Space of fMRI Selectivity Profiles." *NeuroImage* 50 (3): 1085–98.
<https://doi.org/10.1016/j.neuroimage.2009.12.106>.
- Madsen, Jens, and Lucas C Parra. 2022. "Cognitive Processing of a Common Stimulus Synchronizes Brains, Hearts, and Eyes." *PNAS Nexus* 1 (1): pgac020.
<https://doi.org/10.1093/pnasnexus/pgac020>.
- Magliano, Joseph P., and Jeffrey M. Zacks. 2011. "The Impact of Continuity Editing in Narrative Film on Event Segmentation: Cognitive Science." *Cognitive Science* 35 (8): 1489–1517.
<https://doi.org/10.1111/j.1551-6709.2011.01202.x>.
- Margulies, Daniel S., Satrajit S. Ghosh, Alexandros Goulas, Marcel Falkiewicz, Julia M. Huntenburg, Georg Langs, Gleb Bezgin, et al. 2016. "Situating the Default-Mode Network along a Principal Gradient of Macroscale Cortical Organization." *Proceedings of the National Academy of Sciences* 113 (44): 12574–79.
<https://doi.org/10.1073/pnas.1608282113>.
- Markello, Ross D., Justine Y. Hansen, Zhen-Qi Liu, Vincent Bazinet, Golia Shafiei, Laura E. Suárez, Nadia Blostein, et al. 2022. "Neuromaps: Structural and Functional Interpretation of Brain Maps." *Nature Methods* 19 (11): 1472–79.
<https://doi.org/10.1038/s41592-022-01625-w>.
- Wang, Xiao-Jing. 2020. "Macroscopic Gradients of Synaptic Excitation and Inhibition in the Neocortex." *Nature Reviews Neuroscience* 21 (3): 169–78.
<https://doi.org/10.1038/s41583-020-0262-x>.
- Wolpert, Daniel M., Susan J. Goodbody, and Masud Husain. 1998. "Maintaining Internal Representations: The Role of the Human Superior Parietal Lobe." *Nature Neuroscience* 1 (6): 529–33. <https://doi.org/10.1038/2245>.
- Wu, Chia-Chien, Hsueh-Cheng Wang, and Marc Pomplun. 2014. "The Roles of Scene Gist and Spatial Dependency among Objects in the Semantic Guidance of Attention in Real-World Scenes." *Vision Research* 105 (December): 10–20.
<https://doi.org/10.1016/j.visres.2014.08.019>.
- Yeo, Thomas, Fenna M. Krienen, Jorge Sepulcre, Mert R. Sabuncu, Danial Lashkari, Marisa Hollinshead, Joshua L. Roffman, et al. 2011. "The Organization of the Human Cerebral Cortex Estimated by Intrinsic Functional Connectivity." *Journal of Neurophysiology* 106 (3): 1125–65. <https://doi.org/10.1152/jn.00338.2011>.
- Zheng, Jie, Andrea Gómez Palacio Schjetnan, Mar Yebra, Clayton Mosher, Suneil Kalia, Taufik A. Valiante, Adam N. Mamelak, Gabriel Kreiman, and Ueli Rutishauser. 2021. "Cognitive Boundary Signals in the Human Medial Temporal Lobe Shape Episodic Memory Representation." Preprint. Neuroscience. <https://doi.org/10.1101/2021.01.16.426538>.

REVIEWERS' COMMENTS

Reviewer #1 (Remarks to the Author):

I appreciate your careful revision of the article! I have no new comments

Reviewer #2 (Remarks to the Author):

The authors have thoroughly addressed all of my concerns.

Reviewer #3 (Remarks to the Author):

I thank the authors for addressing my comments thoroughly. I feel this manuscript will make a valuable contribution to the literature.